# A foundation for exact binarized morphological neural networks

**Theodore Aouad**
Universite Paris-Saclay, CentraleSupelec, Inria, CVN
3 rue Joliot Curie, Gif-sur-Yvette, France
theodore.aouad@centralesupelec.fr

**Hugues Talbot**
Universite Paris-Saclay, CentraleSupelec, Inria, CVN
3 rue Joliot Curie, Gif-sur-Yvette, France
hugues.tablot@centralesupelec.fr

## Abstract

Training and running deep neural networks (NNs) often demands significant computation and energy-intensive specialized hardware (e.g. GPU, TPU...). One way to reduce the computation and power cost is to use binary weight NNs, but these are hard to train because their derivatives have a non-smooth gradient. We present a model based on Mathematical Morphology (MM), which can binarize ConvNets without loss of performance under certain conditions, but these conditions may not be easy to satisfy in real-world scenarios. To ameliorate this, we propose two new approximation methods and develop a robust theoretical framework for ConvNets binarization using MM. We also propose several regularization losses to improve the optimization. We empirically show that our model can learn a complex morphological network, and explore its performance on a classification task.

## 1 Introduction

Binary weight neural networks (BWNNs) are attractive because they can provide powerful machine learning solutions with much less storage, computation, and energy consumption than conventional networks [21]. Several methods, such as BinaryConnect [5], DoReFa-Net [27], and XNOR-Net [21], have shown good to excellent results in a variety of applications. These networks usually use the sign function to binarize the weights in the forward pass. They must use a special gradient function, like the Straight-Through Estimator (STE) [3], to overcome the zero gradient problem of the sign function during the backward pass. However, this estimator, while effective in practice, lacks a solid theoretical basis, suggesting the need for a different approach. Some methods avoid using the STE by first training a floating-point neural network and then binarizing it afterwards [12, 8]. However, this approach may lead to approximate binarization and a drop in performance. In this paper, we present a new approach that uses the concepts of Mathematical Morphology (MM) [23] to overcome the drawbacks of existing methods. MM, based on modern set theory and complete lattices, offers a non-linear mathematical framework for image processing. Its basic operators, erosion and dilation, are equivalent to thresholded convolutions [14], underscoring a link between MM and deep learning. Combining these fields can improve the efficiency and results of morphological operations while enhancing our knowledge of deep learning [1]. Recent works on morphological neural networks have explored learning operators and structuring elements using various approaches, such as the max-plus definition [19, 6] and differentiable approximations [24, 18, 9]. However, these methods primarily focus on learning gray-scale MM operators and have not focused on NN binarization.

Workshop on Advancing Neural Network Training at 37th Conference on Neural Information Processing Systems (WANT@NeurIPS 2023).

The Binary Morphological Neural Network (BiMoNN) [2] proposed a well-defined mathematical method that can binarize NN weights without performance loss under certain conditions. However, it is limited to binary inputs and can learn only one filter per layer, limiting the design of modern architectures. In this paper, we improve the BiMoNN framework to overcome these limitations and introduce a new model that can learn any sequence of morphological operators while achieving complete binarization in all cases. We also introduce novel regularization techniques to encourage our model to become morphological. By combining MM concepts with deep learning, our work establishes the basis for a more robust and theoretically sound framework for NN binarization. Our contributions in this work are as follows:

- In §2, we refine the BiMoNN theoretical framework and generalize it to any kind of gray-scale / RGB inputs. We introduce two new layers analogous to the dense and convolutional layers, enabling the transposition of modern architectures.
- In §3, we present a well-defined mathematical binarization method based on MM that works seamlessly with standard frameworks and tools. We also propose two new approximate binarization techniques to deal with the cases where exact binarization is not possible.
- In §4, we introduce three applicable regularization losses, and a fourth whose value is mostly theoretical due to its long computation time.
- In §5, we evaluate the capacity of the binarized BiMoNN to learn complex morphological pipelines without performance loss. Additionally, we investigate its behavior on the MNIST [16] classification task, and the behavior of the introduced regularization techniques.

All propositions are proven in Appendix A. Our code is publicly available at `https://github.com/TheodoreAouad/WANT2023`.

## 2 General Binary Morphological Neural Network

We build on the BiMoNN framework [2] and propose a new Binary Structuring Element (BiSE) neuron. We show how it is morphologically equivalent to binary images by using the notion of almost binary images. We also propose the BiSE Layer (BiSEL) that can learn multiple filters per layer, which is similar to a convolutional layer; and the DenseLUI, a dense layer that can be binarized. In §2.1 and §2.2, we consider binary inputs only. We generalize to any types of inputs in §2.3.

### 2.1 Binary Structuring Element neuron

Let $D$ be the dimension of the image (usually $D = 2$ or $3$). We denote $\Omega_I \subset \mathbb{Z}^D$ the support of the images, and $\Omega$ the support of the weights kernel. For the remainder of the paper, we assume $S \subset \Omega$ and $S \neq \emptyset$. For a set $X \subset \mathbb{Z}^D$, we denote its indicator function $\mathbb{1}_X : \mathbb{Z}^D \mapsto \mathbb{R}$ such that $\mathbb{1}_X(i) = 1$ if $i \in X$, else $\mathbb{1}_X(i) = 0$. We denote the set of its subsets $\mathcal{P}(X)$. We denote the convolution by $\circledast$. We denote $[\cdot]_+ := \max(\cdot, 0)$ and $[\cdot]_- := \min(\cdot, 0)$.

**Definition 2.1** (BiSE neuron). Let $(W, B)$ be a set of reparametrization functions and $(\omega, \beta, p)$ a set of learnable parameters. Let $\xi$ be a smooth threshold activation (e.g. normalized $\tanh$: $\frac{1}{2}\tanh(\cdot) + \frac{1}{2}$). Then the *Binary Structuring Element neuron* (BiSE) is:

$$\chi : \mathbf{x} \in [0,1]^{\Omega_I} \mapsto \xi\left(p\Big[\mathbf{x} \circledast W(\omega) - B(\beta)\Big]\right) \in [0,1]^{\Omega_I}. \tag{1}$$

The reparametrization functions $(W, B)$ are hyperparameters. The BiSE neuron is a convolution operator with weights and biases that are reparametrized (see §2.4), a smooth threshold activation and a scaling factor $p$ that can invert the output if negative. We introduce the following *almost binary* image representation, to handle images that are not exactly binary and to be able to apply gradient descent optimization.

**Definition 2.2** (Almost Binary Image). The set of almost binary images of parameter $\delta$ is denoted $\mathcal{I}(\delta)$. We say an image $\mathbf{I} \in [0,1]^{\Omega_I}$ is *almost binary* and define $X_{\mathbf{I}}$ its *associated binary image* if:

$$\exists \delta \in \left]0, \frac{1}{2}\right], \ \mathbf{I}(\Omega_I) \cap \left]\frac{1}{2} - \delta, \frac{1}{2} + \delta\right[ = \emptyset, \tag{2}$$

$$X_{\mathbf{I}} := \left(\mathbf{I} > \frac{1}{2}\right). \tag{3}$$

## 2.2 Morphological Equivalence

We now express the conditions under which a BiSE neuron can be seen as a morphological operator.

**Theorem 2.3** (Dilation - Erosion Equivalence). *For a given structuring element (SE) $S \subset \Omega$, and an almost binary parameter $\delta \in ]0, \frac{1}{2}]$, a set of reparametrized weights $\mathbf{W} \in \mathbb{R}^{\Omega}$ and bias $b \in \mathbb{R}$, we define:*

$$L_{\oplus}(\mathbf{W}, S) := \sum_{k \in \Omega \setminus S} [\mathbf{W}_k]_+ + \left(\frac{1}{2} - \delta\right) \sum_{s \in S} [\mathbf{W}_s]_+ \tag{4}$$

$$U_{\oplus}(\mathbf{W}, S) := \left(\frac{1}{2} + \delta\right) \min_{s \in S} \mathbf{W}_s + \sum_{k \in \Omega} [\mathbf{W}_k]_- \tag{5}$$

$$U_{\ominus}(\mathbf{W}, S) := \sum_{k \in \Omega} \mathbf{W}_k - L_{\oplus}(\mathbf{W}, S) \tag{6}$$

$$\tag{7}$$

$$L_{\ominus}(\mathbf{W}, S) := \sum_{k \in \Omega} \mathbf{W}_k - U_{\oplus}(\mathbf{W}, S) \tag{8}$$

*Let $\psi \in \{\oplus, \ominus\}$ be a dilation or erosion. Then:*

$$\forall I \in \mathcal{I}(\delta) , \ \psi_S\left(\mathbf{I} > \frac{1}{2}\right) = \left(\mathbf{I} \circledast W > b\right) \Leftrightarrow L_{\psi}(\mathbf{W}, S) \leq b < U_{\psi}(\mathbf{W}, S). \tag{9}$$

*In this case, $\forall s \in S, \mathbf{W}_s \geq 0$ and $b \geq 0$ and we say that a BiSE $\chi$ with weights $W(\omega) = \mathbf{W}$ and $B(\beta) = b$ is **activated**. If $\psi = \oplus$, then $B(\beta) \leq \frac{1}{2} \sum_{k \in \Omega} W(\omega)_k$. If $\psi = \ominus$, then $B(\beta) \geq \frac{1}{2} \sum_{k \in \Omega} W(\omega)_k$. For any almost binary image $\mathbf{I} \in \mathcal{I}(\delta)$, $\chi(\mathbf{I}) \in \mathcal{I}(\delta_{out})$ is almost binary with known parameter $\delta_{out}$. Finally*

$$\forall \mathbf{I} \in \mathcal{I}(\delta), \left(\chi(\mathbf{I}) > \frac{1}{2}\right) = \psi_S\left(\mathbf{I} > \frac{1}{2}\right). \tag{10}$$

$$
\begin{array}{ccc}
\mathbf{I} & \xrightarrow{\ \chi\ } & \chi(\mathbf{I}) \\
{\scriptstyle \cdot > \frac{1}{2}} \downarrow & & \downarrow {\scriptstyle \cdot > \frac{1}{2}} \\
X_{\mathbf{I}} & \xrightarrow{\ \psi_S\ } & X_{\chi(\mathbf{I})}
\end{array}
$$

This theorem states that if a BiSE is activated, it transforms an almost binary inputs into an almost binary outputs with known $\delta_{out}$. Moreover, if we threshold the input and output with respect to $\frac{1}{2}$, it is equivalent to applying the corresponding morphological operation, exhibiting a weak commutativity property between thresholding and convolution. Further, equation (10) shows that if a BiSE is activated for operation $\psi$, performing the BiSE operation in $\mathcal{I}(\delta)$ is equivalent to performing the binary morphological operation $\psi$ in the binary space $\mathcal{P}(S)$. This presents a natural framework for binarization (see §3).

## 2.3 Binary Morphological Neural Network

Our objective is to build a binarizable neural network. We now define binarizable neural layers based on the BiSE neuron. By combining these layers, we can create flexible architectures tailored to the desired task. As mentioned earlier, the BiSE neuron resembles a convolution operation. However, a single convolution is insufficient to create a morphological layer. In this context, we explain how we handle multiple channels. We observe that the BiSE neuron can be used to define a layer that learns the intersection or union of multiple binary images $\mathbf{x}_1, ..., \mathbf{x}_n \in \mathcal{I}(\delta)$. For example, their union can be expressed as the dilation of the 3D image $\mathbf{x} := (\mathbf{x}_1, ..., \mathbf{x}_n) \in \mathcal{I}(\delta) \subset \left([0, 1]^{\Omega_I}\right)^n$ with a tubular SE applied solely across the dimension of depth. Therefore, we define the Layer Union Intersection (LUI) as a special case of the BiSE layer, with weights restricted to deep-wise shape. It is analogous

to a $1 \times 1$ convolution unit. A LUI layer can learn any intersection or union of any number of almost binary inputs. By combining BiSE neurons and LUI layers, we can learn morphological operators and aggregate them as unions or intersections.

**Definition 2.4** (BiSEL). A BiSEL (BiSE Layer) is the combination of multiple BiSE and multiple LUI. Let $(\chi_{n,k})_{n,k}$ be $N * K$ BiSE and $(\text{LUI}_k)_k$ be $K$ LUI. Then we define a BiSEL as:

$$\phi: \quad \mathbf{x} \in \left([0,1]^{\Omega_I}\right)^N \mapsto \left(\text{LUI}_k\left[\left(\chi_{n,k}(\mathbf{x}_n)\right)_n\right]\right)_k. \tag{11}$$

The BiSEL mimics a convolutional layer. In conventional ConvNets, to process multiple input channels, a separate filter is applied to each channel, and their outputs are summed to create one output channel. In the case of BiSEL, instead of summing the results of each filter, we perform a union or intersection operation (see Figure 1).

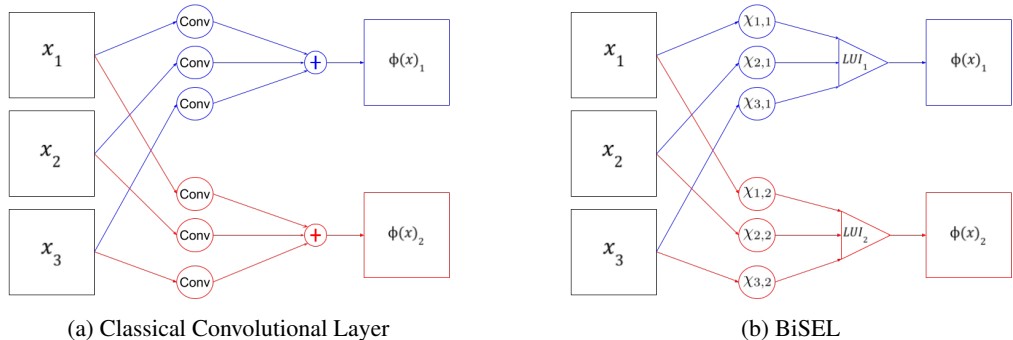

(a) Classical Convolutional Layer

(b) BiSEL

Figure 1: BiSEL vs Conv Layer. Input $\mathbf{x}$ with 3 channels. Output $\phi(\mathbf{x})$ with 2 channels.

**DenseLUI**  the LUI layer is similar to a $1 \times 1$ convolution, which is equivalent to a fully connected layer. Given an input vector $\mathbf{x} \in \mathbb{R}^n$, we can apply the LUI layer to the reshaped input $\hat{\mathbf{x}} \in \mathbb{R}^{n \times 1 \times 1}$ treating it as a 2D image with width and length of $1$ and $n$ channels, respectively. Therefore, we can utilize the BiSEL to create binarizable fully connected layers.

**Gray-Scale / RGB Inputs**  Up until now, all inputs were assumed binary. We extend these definitions to gray-scale and RGB images. The main idea is to separate an input channel $\mathbf{I}_c \in \mathbb{R}^{\Omega_I}$ into its set of upper level-sets to come back to binary inputs:

$$\{(\mathbf{I}_c \geq \tau) \mid \tau \in \mathbb{R}\}. \tag{12}$$

Considering all possible values of $\tau$ from a continuous image would result in an excessive number of level-sets to process. Alternatively, we can define a finite set of values for $\tau$ in advance. Subsequently, each channel of an image $\mathbf{I} \in \mathbb{R}^{c \times w \times l}$ is separated into its corresponding level-sets, and these level-sets are provided as additional channels. If we have $N$ values for level-set, the resulting input is a binary image $I_{\mathbb{B}} \in \{0,1\}^{(N \cdot c) \times w \times l}$.

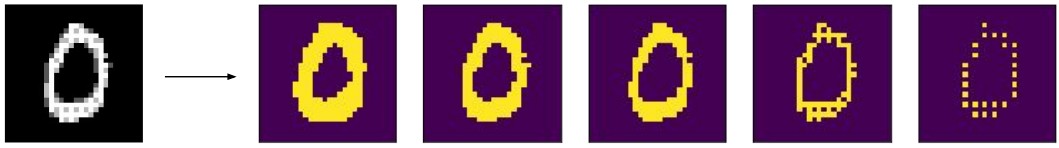

Figure 2: Gray to level-set for 5 different values, generating 5 input channels.

We have introduced two types of binarizable layers: the DenseLUI, which is similar to a fully connected layer, and the BiSEL, which resembles a convolutional layer. By combining these layers, we can create a Binary Morphological Neural Network (BiMoNN), which encompasses various architectures suited for different tasks.

## 2.4 Training considerations

The BiMoNN ($\Gamma_\mathbb{R}$) is fully differentiable. If $\mathcal{L}$ is a differentiable loss function, given a dataset of $N$ labeled images $\{(\mathbf{x}_i, \mathbf{y}_i)\}$, we minimize the error $\frac{1}{N} \sum_{i=1}^{N} \mathcal{L}(\Gamma_\mathbb{R}(\mathbf{x}_i), \mathbf{y}_i)$ using a gradient descent algorithm (like Adam [13]). The gradients are computed with the backpropagation algorithm [22]. The binarization scheme, which is defined in the next section, is applied post-training or during training to measure the model's evolution.

Our objective is to reach the set of activable weights and bias. Theorem 2.3 indicates that we only have to look at positive parameters. We can enforce them to be positive by setting $W$ and $B$ as the softplus function. Other reparametrizations are proposed in Appendix B.

$$B(\cdot) = W(\cdot) := f^+(\cdot) := \log(1 + \exp(\cdot)). \tag{13}$$

# 3 Binarization

Binarization of a neural network involves converting the real-valued weights and activations, typically stored as 32-bit floats, into binary variables represented as 1-bit booleans. Most binary NN use variables in $\{-1, 1\}$ [26], which are not ideal for learning morphological operations. We instead utilize $\{0, 1\}$. BiMoNNs inherently correspond to binarized networks, making the BiSE neuron a natural framework for binarization when dealing with binary inputs. If a specialized hardware tailored for morphological operations is available, it can offer significant improvements in efficiency and performance, facilitating the binarization process. Alternatively, dilation and erosion can be expressed using binary-weighted thresholded convolutions. In our approach, binarization occurs after the training phase. We present two types of binarization for the BiSE neuron: the exact method (as introduced in [2]), when the BiSE neuron is activated, and two novel approximated methods. Then, we sequentially binarize the entire network.

## 3.1 Exact BiSE Binarization

The real-value operations performed by an activated Binary Structuring Element (BiSE) in the almost binary space can be replaced with binary morphological operations on the binary space after thresholding at 0.5, without sacrificing performance (as per Theorem 2.3). To determine if a BiSE is activated and which operation it corresponds to, we introduce Proposition 3.1, which provides a linear complexity method for extraction.

**Proposition 3.1** (Linear Check)**.** *Let us assume the BiSE of weights $W(\omega)$ and $B(\beta)$ is activated for $\psi$ with SE $S \subset \Omega$ for almost binary images $\mathcal{I}(\delta)$. Then $S = (W(\omega) > \tau_\psi)$ with*

$$\tau_\oplus := \frac{1}{\frac{1}{2} + \delta} \Big( B(\beta) - \sum_{W(\omega)} [w]_- \Big), \tag{14}$$

$$\tau_\ominus := \frac{1}{\frac{1}{2} + \delta} \Big( \sum_{W(\omega)} [w]_+ - B(\beta) \Big). \tag{15}$$

Given a BiSE neuron $\chi$ and an almost binary output in $\mathcal{I}(\delta)$, we check if $\chi$ is activated for $S_\oplus$ or $S_\ominus$, where $S_\oplus = (W(\omega) > \tau_\oplus)$ and $S_\ominus = (W(\omega) > \tau_\ominus)$. If yes, we binarize by replacing $\chi$ with the corresponding morphological operator. If no, we use proposition 3.1 to confirm that $\chi$ is not activated, and we approximate the binarization using the methods in section 3.2. The exact method requires only the computation of $L_\oplus, U_\oplus, L_\ominus, U_\ominus$ and at most $\mathcal{O}(|\Omega_S|)$ operations.

## 3.2 Approximate BiSE Binarization

In practice, BiSE are not always activated, necessitating an approximate binarization method. Let $(\widehat{\mathbf{W}}, \widehat{b}, \widehat{p}) := (\mathbf{W}(\hat{\omega}), B(\hat{\beta}), \widehat{p})$ be the learned reparametrized parameters.

### 3.2.1 Projection onto activable parameters

To find the closest morphological operation, we minimize the Euclidean distance $d$ for a given $\psi \in \{\oplus, \ominus\}$ and the set $A_{\psi,S}$ of activable parameters:

$$A_{\psi,S} := \left\{ (\mathbf{W}, b) \in \mathbb{R}^\Omega \times \mathbb{R} \mid L_\psi(\mathbf{W}, S) < b < U_\psi(\mathbf{W}, S) \right\}, \tag{16}$$

$$\underset{S \subset \Omega, \psi \in \{\oplus, \ominus\}}{\text{minimize}} \quad d\left( (\widehat{\mathbf{W}}, \widehat{b}), A_{\psi,S} \right). \tag{17}$$

For each set $A_\psi(S)$ corresponding to a possible morphological operation, we find the smallest distance for each $(S, \psi)$ and apply complementation if $\hat{p} < 0$. The following proposition results in a linear search instead of exponential.

**Proposition 3.2.** *If $S^*, \psi^*$ minimize $d\left( (\widehat{\mathbf{W}}, \widehat{b}), A_{\psi,S} \right)$, then $S^* = (\widehat{\mathbf{W}} \geq \min_{S^*} \widehat{\mathbf{W}}_s)$.*

Proposition 3.1 guarantees that the SE is a set of thresholded weights when the weights are activated. Proposition 3.2 ensures that this property is preserved for the optimal SE even when the weights are not activated. Thus, we only need to compute the distance to all possible sets of thresholded weights, denoted as $S_k := (\widehat{\mathbf{W}} \geq \widehat{\mathbf{W}}_k)$, and select the smallest distance. To compute the distance for $\psi$ and $S$ fixed, we solve the following optimization problem:

$$\underset{(\mathbf{W}, b) \in \mathbb{R}_+^\Omega \times \mathbb{R}}{\text{minimize}} \quad \frac{1}{2} \sum_{i \in \Omega} (\mathbf{W}_i - \widehat{\mathbf{W}}_i)^2 + \frac{1}{2}(b - \widehat{b})^2 \quad \text{subject to} \quad \begin{cases} L_\psi(\mathbf{W}, S) - b \leq 0 \\ b - U_\psi(\mathbf{W}, S) \leq 0 \end{cases}. \tag{18}$$

When the initial weights $\widehat{\mathbf{W}}$ are positive, the projected weights are also positive. Consequently, the constraints can be rewritten for dilation and erosion as follows:

$$\oplus \begin{cases} \sum_{k \in \Omega \setminus S} \mathbf{W}_k - b \leq 0 & (19) \\ \forall s \in S, b \leq \mathbf{W}_s & (20) \\ \forall k \in \Omega \setminus S, -\mathbf{W}_k \leq 0 & (21) \end{cases} \qquad \ominus \begin{cases} b - \sum_{s \in S} \mathbf{W}_s \leq 0 & (22) \\ \forall s \in S, \sum_{i \in \Omega} \mathbf{W}_i - \mathbf{W}_s \leq b & (23) \\ \forall k \in \Omega \setminus S, -\mathbf{W}_k \leq 0. & (24) \end{cases}$$

These new constraints are then linear. By utilizing the positive reparametrization technique (as described in §2.4), we can compute the distance to any activable parameter set $A_{\psi,S}$ by solving a QP problem, using the OSQP solver [25]. This needs to be done for all possible sets of thresholds $S_k$, of which there are $|\Omega|$. For a single layer with 4096 input and output neurons, the computation would take up to 28 days (on a Intel(R) Xeon(R) W-2265 CPU, 3.50GHz). Future work may improve this by finding an analytically computable form for efficient distributed computing. In the mean time, we now introduce an efficient approximation technique.

### 3.2.2 Projection onto constant weights

We use a similar technique to [17]. First, we define $\widetilde{A}_S$ the set of constant weights over $S$, and replace $d((\widehat{\mathbf{W}}, \widehat{b}), A_{\psi,S})$ with $d(\widehat{\mathbf{W}}, \widetilde{A}_S)$.

$$\widetilde{A}(S) := \{\theta \cdot \mathbb{1}_S \mid \theta > 0\}, \tag{25}$$

$$d(\widehat{\mathbf{W}}, \widetilde{A}_S) = \sum_{i \in \Omega} \widehat{\mathbf{W}}_i^2 - \frac{1}{|S|} \left( \sum_{s \in S} \widehat{\mathbf{W}}_s \right)^2. \tag{26}$$

Similarly to proposition 3.2, we can prove that the optimal $S^*$ is a set of thresholded weights. The analytical form in 26 allows for the efficient computation of $S^*$ in distributed systems, significantly faster than the first method: for a layer with 500 input and output channels, this step takes less than 3 seconds. Once $S^*$ is obtained, the bias term helps determine $\psi \in \{\oplus, \ominus\}$, based on Theorem 2.3. If $\widehat{b} > \sum_{\widehat{\mathbf{W}}} w/2$, the operation is an erosion; otherwise, it is a dilation.

### 3.3 BiMoNN binarization

The core of the successive binarization is Theorem 2.3. To simplify, we suppose that a BiMoNN is a succession of BiSE. If the first BiSE is activated, then with a binary input (e.g. almost binary with

$\delta_0 = \frac{1}{2}$), the output is almost binary of known parameter $\delta_1$. If the next BiSE is activated for the parameter $\delta_1$, its output is also activated for parameter $\delta_2$, and so on. Hence, every BiSE operation on the almost binary space is equivalent to the morphological operation on the binary space. In case one BiSE is not activated, its output is not on the almost binary space, breaking the exact equivalence between BiSE and morphology. We apply an approximate method (projection §3.2.1 for a small network, otherwise our fast projection §3.2.2), and assume that the input for the next BiSE is binary.

## 4  Morphological Regularization

The binarization schema, as described in §3, is separate from the training process. During standard loss optimization with classical gradient descent, there is no explicit constraint that makes the network behave morphologically. Consequently, the network may not tend to a morphological operation: BiSE operators may not be activated and the weights may stay far from their respective projection space, resulting in potential errors in the approximate binarization process compared to the floating-point operator. To encourage the network to exhibit a more morphological behavior, we propose the inclusion of a regularization term in the loss, denoted as $\mathcal{L}_{\text{morpho}}$. The loss function now becomes:

$$\mathcal{L} = \mathcal{L}_{\text{data}} + c \cdot \mathcal{L}_{\text{morpho}}. \tag{27}$$

$\mathcal{L}_{\text{data}}$ represents the loss used for the data-driven task (e.g. Cross-Entropy Loss for classification), and $c$ is the hyperparameter controlling the strength of the regularization term. To define the regularization term, we reuse the two approximate binarization techniques defined in section 3.2.

### 4.1  Regularization onto activable parameters

A first idea is to reduce the distance to the closest set $A_{\psi,S}$ defined in (16) for each BiSE neuron. Let $(\mathbf{W}, b) := (W(\omega), B(\beta))$ be the weights in a given iteration:

$$\mathcal{L}_{\text{morpho}} = \mathcal{L}_{\text{acti}} := \min_{S,\psi} d\bigg( (\mathbf{W}, b), A_{\psi,S} \bigg). \tag{28}$$

To do this, we must compute this distance in a differentiable fashion. We proceed in two steps: first, we compute the optimal $(S^*, \psi^*)$ as in §3.1, by checking all possible thresholded set of weights and solving the corresponding QP with OSQP, with the Lagrangian dual method. This yields the optimal $(\mathbf{W}^*, b^*)$ as well as the Lagrangian dual values, from which we deduce the differentiable form of the distance. More details are given in Appendix D. However, the computational burden described in §3.1 persists: the first step of computing $S^*$ is too long in practice.

### 4.2  Regularization onto constant set

Instead of trying to enforce a fully morphological network, we can encourage the weights to stay in the set of constant weights $\widetilde{A}_S$ defined in (25). We proceed in two steps: we find the best $S^*$ in the same way as in §3.2.2, i.e by computing the distance $d(\widetilde{A}_S, \mathbf{W})$ defined in (26) for all set of thresholded weights, and selecting the smallest distance. Then, a differentiable expression for the distance is:

$$\mathcal{L}_{\text{morpho}} = \mathcal{L}_{\text{exact}} := \sum_{i \in \Omega} \mathbf{W}_i - \frac{1}{|S^*|} \bigg( \sum_{s \in S^*} \mathbf{W}_s \bigg)^2 \quad . \tag{29}$$

Computing the distance for all set of thresholded weights still takes significant time, and in our experimentation, this slows the training by up to $80\times$. Instead, we propose to use the technique introduced in [17]. If we assume that the weights follow a uniform or normal distribution, we can approximate the optimal $S^* \simeq (\mathbf{W} > \tau)$ with the following thresholds:

$$\tau_u := \frac{2}{3}\bigg( \frac{1}{|\Omega|} \sum_{i \in \Omega} \mathbf{W}_i \bigg), \quad S_u := (\mathbf{W} > \tau_u), \tag{30}$$

$$\tau_n := \frac{3}{4}\bigg( \frac{1}{|\Omega|} \sum_{i \in \Omega} \mathbf{W}_i \bigg), \quad S_n := (\mathbf{W} > \tau_n). \tag{31}$$

From this, we can define two regularization loss which can be computed without slowing the training down:

$$\mathcal{L}_{\text{morpho}} := \mathcal{L}_{\text{unif}} := \sum_{i \in \Omega} \mathbf{W}_i - \frac{1}{S_u} \left( \sum_{s \in S_u} \mathbf{W}_s \right)^2, \tag{32}$$

$$\mathcal{L}_{\text{morpho}} := \mathcal{L}_{\text{normal}} := \sum_{i \in \Omega} \mathbf{W}_i - \frac{1}{S_n} \left( \sum_{s \in S_n} \mathbf{W}_s \right)^2. \tag{33}$$

## 5 Experiments

In this section, we empirically validate the capabilities of BiMoNNs in learning a binarized morphological pipeline through a denoising task, without the need for regularization. We also evaluate the model and regularization techniques on the MNIST classification task.

**Binary Denoising**    We generate a second dataset to evaluate the denoising capacity of BiMoNNs. The target images in this dataset consist of randomly-oriented segments with width $h$, with added Bernoulli noise. To filter these images, an MM expert would use a union of opening operations, where the SEs are segments with width 1 and angle one of $(0°, 90°, -45°)$. The SEs should be longer than the noise and shorter than the smallest stick in the target image (usually a length of 5). Examples are given in Figure 3a. Our architecture uses two consecutive BiSEL layer of kernel size 5 and 3 hidden channels (see Figure 3b). We optimize the MSE Loss, with an initial learning rate of 0.01. We train over 6000 iterations and halve the learning rate after 700 iterations with non-diminishing loss. We stop once the loss does not decrease after 2100 iterations. We employ a positive reparametrization for the bias and a dual reparametrization for the weights, and binarize with the projection onto activable parameters. The network achieves excellent denoising performance, with a DICE score of 97.5%. The remaining 2.5% discrepancy is due to artifacts between the sticks that cannot be denoised using an opening operation. The binarized network (Figure 3b) accurately learns the intersection of three openings (which remains an opening) the same way a human expert would combine such operators to achieve the denoising task optimally. Additionally, 4 out of the 6 BiSE are activated during the process. This experiment shows that our network can learn accurate and interpretable composition of morphological operators.

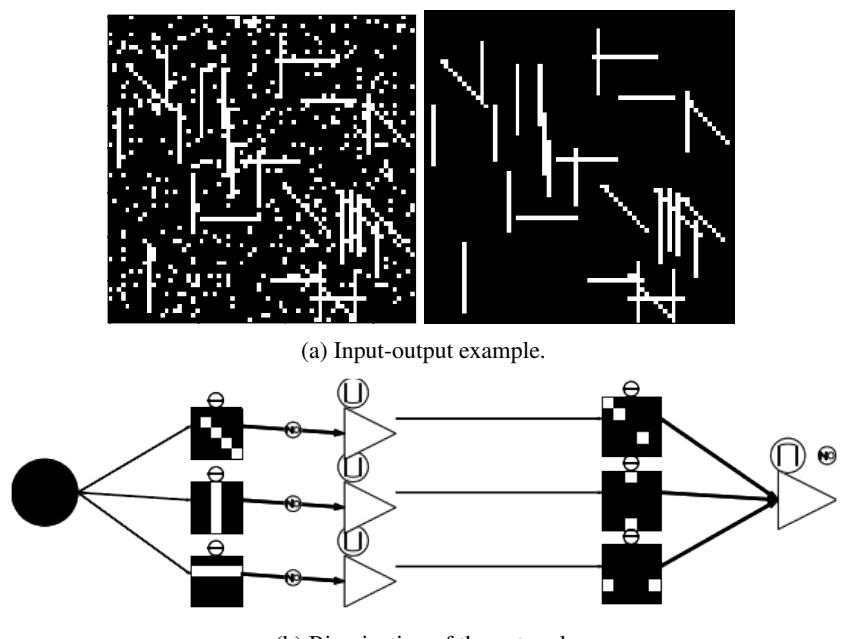

(a) Input-output example.

(b) Binarization of the network

Figure 3: Binary denoising experiment.

**Classification**   We conduct classification experiments on the MNIST dataset [16]. All images are thresholded at 128. Our BiMoNN model comprises one hidden DenseLUI layer with 4096 neurons. To handle the large number of parameters, we adopt the fast projection defined in §3.2. We compare the classification accuracy of our float and binarized models against the SOTA and baseline models with fully connected layers and 1D batch normalization [11], employing $\frac{1}{2}(\tanh(\cdot) + 1)$ as the activation layer. The accuracy results are summarized in Table 1. In our framework, binarizing the weights also entails binarizing the activations. Consequently, binarizing the last layer would yield binary decision outputs for each output neuron, possibly leading to multiple labels with a score of 1. To overcome this issue, we refrain from binarizing the last layer, thus retaining the real-valued activations. This decision affects a negligible proportion of parameters ($\simeq 0.1\%$). In traditional classification neural networks, the softmax activation is commonly used at the end of the last layer to produce the final probability distribution over the classes. However, in the BiMoNN architecture, we utilize the same activation function as the hidden layers, which is the normalized `tanh`. Additionally, we compare the performance of our BiMoNN model when replacing the last normalized `tanh` activation with a softmax layer. When using the normalized `tanh`, $\mathcal{L}_{data}$ is the Binary Cross-Entropy loss $\mathcal{L}_{BCE}$ , and when using the softmax, we use the Cross-Entropy loss $\mathcal{L}_{CE}$:

$$\mathcal{L}_{BCE}(\hat{\mathbf{y}}_i, \mathbf{y}_i^*) \coloneqq \sum_{c=0}^{9} \mathbf{y}_i^* \log(\hat{\mathbf{y}}_i) + (1 - \mathbf{y}_i^*) \log(1 - \hat{\mathbf{y}}_i), \tag{34}$$

$$\mathcal{L}_{CE}(\hat{\mathbf{y}}_i, \mathbf{y}_i^*) \coloneqq \sum_{c=0}^{9} \mathbf{y}_i^* \log(\hat{\mathbf{y}}_i). \tag{35}$$

We conduct a comprehensive random search to identify the optimal hyperparameter configuration for the Binary Morphological Neural Network (BiMoNN). The hyperparameters explored include the learning rate, last activation function (Softmax layer vs. normalized `tanh`), positive vs no reparametrization. Additionally, we investigate regularization losses, such as no regularization, $\mathcal{L}_{exact}$, $\mathcal{L}_{uni}$, and $\mathcal{L}_{normal}$. If regularization is applied, only positive weight reparametrization is considered. We vary the coefficient $c$ in the regularization loss and explore different batch value starting time for when we start applying regularization during training. For each regularization schema, we select the model with the best binary validation accuracy, and the corresponding results are displayed in Table 1. Detailed hyperparameter configurations and hyperparameters study are provided in Appendix E. Applying the softplus reparametrization to the weights led to a slight increase in the floating-point

Table 1: Accuracy error on test set for MNIST classification, with float error $\mathbb{R}$ and binarized error $\mathbb{B}$.

|      | Architecture | Params | $\mathbb{R}$ | $\mathbb{B}$ |
|------|--------------|--------|------|------|
| Ours | DLUI ($W = \text{Id}$) | 3.3 M | **2.2%** | 90.2% |
|      | DLUI (No Regu) | 3.3 M | 4.6% | 10.1% |
|      | DLUI $\mathcal{L}_{exact}$ | 3.3 M | 4.0% | 7.3% |
|      | DLUI $\mathcal{L}_{unif}$ | 3.3 M | 3.6% | **4.5%** |
|      | DLUI $\mathcal{L}_{normal}$ | 3.3 M | 2.8% | 4.6% |
| SOTA | EP 1fc [15] | 3.3 M | - | 2.8% |
|      | BinConnect [5] | 10 M | - | **1.3%** |
|      | BNN [10] | 10 M | - | 1.4% |
| Float | FC (4096) | 3.3 M | 1.5 % | - |
|      | FC (2048x3) [10] | 10 M | **1.3%** | - |

error (2.2% vs. 2.8%). Similar findings were observed in [20] for non-negative neural networks in a different task. Generally, positive neural networks exhibit lower accuracy but offer enhanced robustness and interpretability [4]. In our case, it significantly improved the binarized results, along with a substantial increase in the rate of activated BiSE neurons, rising from a median of 1.5% to 10% with softplus. Without imposing positivity, the binarized network performed randomly. We analyze the impact of regularization on the performance of the binarized model, which improves as expected. The float accuracy also increases, given that we select the model with the best binary accuracy on validation. Surprisingly, $\mathcal{L}_{unif}$ and $\mathcal{L}_{normal}$ outperform $\mathcal{L}_{exact}$, despite being designed as approximations. This discrepancy might be due to the number of searches performed: $\mathcal{L}_{exact}$ performed only 42 searches, while other configurations went through 100 searches. However, we have not yet achieved parity with the baseline for the float model or reached the state-of-the-art for

the binarized model. With $4.5\%$ error compared to $2.8\%$ for the same number of parameters, this emphasizes the need for improved architecture, better regularization techniques, or exploration of alternative optimization methods.

## 5.1 Discussion

The state-of-the-art BWNN methods commonly rely on the XNOR operator to emulate multiplications. However, this algebraic framework proves unsuitable for morphological operators, as it contradicts the set-theoretical principles of morphological operations. In our experiments, we observed that the BNN operator [10] failed to learn even the simplest dilation operator. Furthermore, the performance of the state-of-the-art method on the denoising task was unsatisfactory, with a DICE coefficient of only approximately 0.3, indicating the need for improved approaches in handling morphological operations. In contrast, our findings reveal that the float BiMoNN exhibits enhanced binarization capabilities when trained to closely approximate a set of morphological operators. As a result, the float BiMoNN naturally acquires morphological properties, leading to a more effective subsequent binarization process. However, when applied to the classification of the MNIST dataset, the resulting float BiMoNN does not retain morphological characteristics, causing a noticeable performance degradation after binarization. To address this issue, we emphasize the importance of positive reparametrization and applying morphological regularization. By incorporating these techniques, we significantly improved the model's overall performance and mitigate the accuracy loss upon binarization. This shows the potential of our proposed BiMoNN framework in leveraging morphological properties and offers insights into the development of more effective BiMoNN models for many and diverse tasks.

## 6 Conclusion

In this paper, we have presented a novel, mathematically justified approach to binarize neural networks using the Binary Morphological Neural Network (BiMoNN), achieved by leveraging mathematical morphology. Our proposed method establishes a direct link between deep learning and mathematical morphology, enabling binarization of a wider set of architectures, without performance loss under specific activation conditions and providing an approximate solution when these conditions are not met. Through our experiments, we have demonstrated the effectiveness of our approach in learning morphological operations and achieving high accuracy in denoising tasks, surpassing state-of-the-art techniques that rely on the straight-through estimator (STE). Furthermore, we proposed and evaluated three practical regularization techniques that aid in converging to a morphological network, showcasing their efficacy in a classification task. Additionally, we introduced a fourth regularization technique that, though promising in theory, currently faces computational challenges. We will adress these shortcoming in future work. Despite promising results, there is still room for enhancing both the floating point and binary modes of our network. As well, diverse architectures, such as incorporating convolution layers, could be explored to further improve the performance and applicability of BiMoNN. Overall, our research lays the foundation for advancing the field of binarized neural networks with a morphological perspective, offering valuable insights into developing more powerful and efficient models for a wide range of tasks.

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

## A  Proofs

### A.1  Proof of theorem 2.3

**Lemma A.1.** *Let*

$$R(\delta) := \left[0, \frac{1}{2} - \delta\right] \bigcup \left[\frac{1}{2} + \delta, 1\right] \tag{36}$$

$$L_1 := \max_{(a_k)_{k \in \Omega_S \setminus S} \in R(\delta)} \sum_{k \in \Omega_S \setminus S} a_k w_k \tag{37}$$

$$L_2 := \max_{(c_k)_{k \in S} \in [0, \frac{1}{2} - \delta]} \sum_{k \in S} c_k w_k \tag{38}$$

$$U_1 := \min_{(a_k)_{k \in \Omega_S \setminus S} \in R(\delta)} \sum_{k \in \Omega_S \setminus S} a_k w_k \tag{39}$$

$$U_2 := \min_{(c_k)_{k \in S} \in R(\delta), \exists j \in S, c_j \geq \frac{1}{2} + \delta} \sum_{k \in S} a_c w_k \tag{40}$$

*The two following propositions are equivalent:*

$$\forall I \in \mathcal{I}(\delta) , \ \left(X_I \oplus S = \{i \in \mathbb{Z}^d \mid I \circledast W(i) > b\}\right) \tag{41}$$

$$L_1 + L_2 \leq b < U_1 + U_2 \tag{42}$$

*Proof.* First we show that $41 \Rightarrow 42$.

We suppose 41.

**Right hand side**

Let $(a_k)_{k \in \Omega_S \setminus S} \in ([0, \frac{1}{2} - \delta] \cup [\frac{1}{2} + \delta, 1])^{\Omega_S \setminus S}$ and $(c_k)_{k \in S} \in ([0, \frac{1}{2} - \delta] \cup [\frac{1}{2} + \delta, 1])^S$ such that $\exists j \in S , \ c_j \geq \frac{1}{2} + \delta$.

Let $I \in [0, 1]^{-\Omega_I}$ be such that $I(-k) = a_k$ if $k \in \Omega_S \setminus S$ and $I(-k) = c_k$ if $k \in S$. Then $I \in \mathcal{I}(\delta)$.

Let $j \in S , \ c_j \geq \frac{1}{2} + \delta$. Then $I(-j) \geq \frac{1}{2} + \delta$ , therefore $-j \in X_I$, and $0 = j - j \in X_I \oplus S$. Therefore,

$$I \circledast W(0) = \sum_{k \in \Omega_S} I(-k) w_k = \sum_{k \in \Omega_S \setminus S} a_k w_k + \sum_{k \in S} c_k w_k > b \tag{43}$$

**Left hand side**

We reason by contradiction. Let us suppose that $\exists (a_k)_{k \in \Omega_S \setminus S} \in R(\delta)^{\Omega_S \setminus S}$ and $\exists (c_k)_{k \in S} \in [0, \frac{1}{2} - \delta]^S$, such that $\sum_{k \in \Omega_S \setminus S} a_k w_k + \sum_{k \in S} a_k w_k > b$. Let $I \in [0, 1]^{-\Omega_I}$ be such that $I(-k) = a_k$ if $k \in \Omega_S \setminus S$ and $I(-k) = c_k$ if $k \in S$. Then $I \in \mathcal{I}(\delta)$. As above, $I \circledast W(0) > b$. But $\forall j \in S, I(-j) \leq \frac{1}{2} - \delta$ and $-j \notin S$. Therefore, $0 \notin X_I \oplus S$. This contradicts $(CD1)$.

$42 \Rightarrow 41$  Let us suppose 42. Let $I \in \mathcal{I}(\delta)$.

- $X_I \oplus S \subset (I \circledast W > b)$

Let $j \in X_I \oplus S$. Then

$$I \circledast W(j) = \sum_{k \in \Omega_S} I(j-k)w_k \tag{44}$$

$$= \sum_{k \in \Omega_S \setminus S} I(j-k)w_k + \sum_{k \in S} I(j-k)w_k \tag{45}$$

Per definition of dilation, $\exists s \in S, \exists x \in X_I$ , $j = s + x$, or in other words, $\exists s \in S$ , $j - s \in X_I$, therefore $\exists s \in S$ , $I(j-s) \geq \frac{1}{2} + \delta$. Using the right hand side of $(CD1)$, we conclude.

- $(I \circledast W > b) \subset X_I \oplus S$

Let $j \in I \circledast W > b$. By contradiction, we suppose that $\forall s \in S, j - s \notin X_I$. Then $\forall s \in S$ , $I(j-s) \leq 0$. Therefore, using the left hand side of $(CD1)$, $I \circledast W(j) = \sum_{k \in \Omega_S \setminus S} I(j-k)w_k + \sum_{k \in S} I(j-k)w_k \leq b$ which contradicts our hypothesis.

$\square$

Now we proof the Theorem 2.3 for dilation:

**Proposition A.2.** *The two following propositions are equivalent:*

$$(\forall I \in \mathcal{I}(\delta)) \ , \ (X_I \oplus S = \{i \in \mathbb{Z}^d \mid I \circledast W(i) > b\}) \tag{46}$$

$$L_1 + L_2 = L_\oplus \leq b < U_\oplus = U_1 + U_2 \tag{47}$$

*If one of them is respected, then $b \geq 0$ and $\forall k \in S, w_k \geq 0$.*

*Proof.* First we show 4 propositions.

**First equality**

$$\max_{(a_k)_{k \in \Omega_S \setminus S} \in R(\delta)^{\Omega_S \setminus S}} \sum_{k \in \Omega_S \setminus S} a_k w_k = \sum_{i \in \Omega_S \setminus S \ , \ w_i \geq 0} w_i \tag{48}$$

Let $(a_k)_{k \in \Omega_S \setminus S} = \underset{(a_k)_{k \in \Omega_S \setminus S} \in R(\delta)^{\Omega_S \setminus S}}{\operatorname{argmax}} \sum_{k \in \Omega_S \setminus S} a_k w_k$. Let $k \in \Omega_S \setminus S$. If $w_k < 0$ , then $a_k = 0$, otherwise replacing only $a_k$ by 0 in the sequence $(a_k)_k$ would result in a higher sum. We apply the same reasoning to show that if $w_k > 0$, then $a_k = 1$. Therefore, $\sum_{k \in \Omega_S \setminus S} a_k w_k = \sum_{k \in \Omega_S \setminus S \ , \ w_k \geq 0} a_k w_k$.

**Second equality**

$$\min_{(a_k)_{k \in \Omega_S \setminus S} \in R(\delta)^{\Omega_S \setminus S}} \sum_{k \in \Omega_S \setminus S} a_k w_k = \sum_{i \in \Omega_S \setminus S \ , \ w_i \leq 0} w_i \tag{49}$$

We apply equation 48 to $\max_{(a_k)_{k \in \Omega_S \setminus S} \in R(\delta)^{\Omega_S \setminus S}} \sum_{k \in \Omega_S \setminus S} (-a_k) w_k$.

**Third equality**

$$\max_{(a_k)_{k \in S} \in [0, \frac{1}{2} - \delta]} \sum_{k \in S} a_k w_k = (\frac{1}{2} - \delta) \sum_{k \in S \ , \ w_k \geq 0} w_k \tag{50}$$

Let $(a_k)_{k \in S} = \underset{(a_k)_{k \in S} \in [0, \frac{1}{2} - \delta]^S}{\operatorname{argmax}} \sum_{k \in S} a_k w_k$. Let $i \in S$. Then $\sum_{k \in S} a_k w_k = \sum_{k \in S \{i\}} a_k w_k + a_i w_i \geq \sum_{k \in S \{i\}} a_k w_k$, therefore $a_i w_i \geq 0$. Therefore, if $w_i < 0$, then $a_i = 0$. If $w_i > 0$, then $a_i = \frac{1}{2} - \delta$ otherwise we could replace $a_i$ by $\frac{1}{2} - \delta$ to get a higher sum. Therefore $\sum_{k \in S} a_k w_k = (\frac{1}{2} - \delta) \sum_{k \in S \ , \ w_k \geq 0} w_k$.

**Fourth equality**

$$\left(\forall i \in S, w_k \geq 0\right) \Rightarrow \min_{(a_k) \in R(\delta), \exists j \in S, a_j \geq \frac{1}{2} + \delta} \sum_{k \in S} a_k w_k = \left(\frac{1}{2} + \delta\right) \min_{k \in S} w_k \tag{51}$$

Let

$$(a_k)_{k \in S} = \operatorname*{argmin}_{(a_k) \in R(\delta)^S, \exists j \in S, a_j \geq (\frac{1}{2} + \delta)} \sum_{k \in S} a_k w_k \tag{52}$$

Let $i \in S$ such that $a_i \leq \frac{1}{2} - \delta$. Then: $\sum_{k \in S} a_k w_k = \sum_{k \in S \setminus \{i\}} a_k w_k + a_i w_i \leq \sum_{k \in S \setminus \{i\}} a_k w_k$ by minimality, then $a_i w_i \leq 0$. as $w_i \geq 0$, we have $a_i = 0$.

Then if $|\{j \in S \mid a_j \geq \frac{1}{2} + \delta\}| > 1$, then we could replace all of them except one by a $0$ to reduce the sum. Therefore, $\exists! j \in S$, $a_j \geq \frac{1}{2} + \delta$. If $\exists j_2 \neq j \in S$, $w_{j_2} < w_j$, we could invert $a_j$ and $a_{j_2}$ to have a lower sum. Therefore, $w_j = \min_{i \in S} w_i$ and $\sum_{k \in S} a_k w_k = (\frac{1}{2} + \delta) \min_{k \in S} w_k$.

$(46 \Rightarrow 47)$

Let us assume $(CD1)$. Then we have the inequality of A.1. Using props 48, 49, 50 and 51, it suffices to show that $\forall i \in S$, $w_i \geq 0$. First we notice that $b \geq 0$. Let $(a_k)_{k \in S} = \operatorname*{argmin}_{(a_k)_{k \in S} \in R(\delta)^S, \exists j \in S, a_j \geq \frac{1}{2} + \delta} \sum_{k \in S} a_k w_k$. Let $k \in S$, $a_j \geq \frac{1}{2} + \delta$. Let $A = \{k \in S \mid w_k < 0\}$ and let us suppose that $A \neq \emptyset$. Then $\sum_{k \neq j \in S, w_k < 0} w_k \geq \sum_{k \in S} a_k w_k > b \geq 0$. We have reached a contradiction. Finally, $\sum_{k \in \Omega_S \setminus S, w_k \leq 0} w_k = \sum_{k \in \Omega_S, w_k \leq 0} w_k$, which concludes this part.

$(47 \Rightarrow 46)$

Let us assume $(CD2)$. Thanks to the preliminary results, it suffices to show that $\forall i \in S$, $w_i \geq 0$.

Using the right hand side of the inequality, we see that $(\frac{1}{2} + \delta) \min_{k \in S} w_k > b \geq 0$. As $\frac{1}{2} + \delta > 0$, this proves the result. $\qquad \square$

With the exact same proof, by interverting the strict inequality, we get:

**Proposition A.3.** *The two following propositions are equivalent:*

$$(\forall I \in \mathcal{I}(\delta)), \ (X_I \oplus S = \{i \in \mathbb{Z}^d \mid I \circledast W(i) \geq b\}) \tag{53}$$

$$L_\oplus < b \leq U_\oplus \tag{54}$$

*If one of them is respected, then $b \geq 0$ and $\forall k \in S, w_k \geq 0$.*

Then we can deduce the results for erosion from the dilation bounds.

**Proposition A.4.** *The two following propositions are equivalent:*

$$(\forall I \in \mathcal{I}(\delta)), \ (X_I \ominus S = \{i \in \mathbb{Z}^d \mid I \circledast W(i) > b\}) \tag{55}$$

$$\sum W - U_\oplus \leq b < \sum W - L_\oplus \tag{56}$$

*Proof.*

$$\forall I \in \mathcal{I}(\delta), X_I \oplus S = \left(I \circledast W \geq \sum_W w - b\right) \tag{57}$$

$$\Leftrightarrow \forall I \in \mathcal{I}(\delta), (X_I^C \ominus S)^C = \left(I \circledast W \geq \sum_W w - b\right) \tag{58}$$

$$\Leftrightarrow \forall I \in \mathcal{I}(\delta), X_I^C \ominus S = \left(I \circledast W < \sum_W w - b\right) \tag{59}$$

$$\Leftrightarrow \forall I \in \mathcal{I}(\delta), X_{1-I}^C \ominus S = \left((1 - I) \circledast W < \sum_W w - b\right) \tag{60}$$

$$\Leftrightarrow \forall I \in \mathcal{I}(\delta), X_I \ominus S = \left(b < I \circledast W\right) \tag{61}$$

Using proposition A.3,

$$57 \Leftrightarrow L_\oplus < \sum_W w - b \leq U_\oplus \tag{62}$$

$$\Leftrightarrow \sum_W w - U_\oplus \leq b < \sum_W w - L_\oplus \tag{63}$$

$\square$

**Corollary A.5** (BiSE duality)**.** *The BiSE of weights $W$ and bias $B$ is activated for $S$ for erosion if and only if the BiSE of weights $W$ and bias $\sum_W w - B$ is activated for $S$ for dilation.*

We also show the inequalities between bias and weights.. First, we show for erosion.

**Proposition A.6.** *Let $S \in \Omega$ and $\left\lceil \frac{\frac{3}{2} - \delta}{\frac{1}{2} + \delta} |S| \right\rceil \geq 3$. If the BiSE of weights $W$ and $B$ is activated for erosion, then $B \geq \frac{1}{2} \sum_W w$*

*Proof.* For erosion

$$\frac{1}{2} \sum_{k \in \Omega_S} w_k = \frac{1}{2} \left[ \sum_{k \in \Omega_S, w_k < 0} w_k + \sum_{k \in S} w_k + \sum_{k \in \Omega_S \setminus S, w_k \geq 0} w_k \right] \tag{64}$$

$$\geq \frac{1}{2} \left[ \sum_{k \in \Omega_S, w_k \geq 0} w_k - \left(\frac{1}{2} + \delta\right) \min_{k \in S} w_k - \left(\frac{1}{2} + \delta\right) \sum_{k \in S} w_k + \sum_{k \in S} w_k + \sum_{k \in \Omega_S \setminus S, w_k \geq 0} w_k \right] \tag{65}$$

$$\geq \frac{1}{2} \left[ 2 \sum_{k \in \Omega_S \setminus S, w_k \geq 0} w_k + \left(\frac{3}{2} - \delta\right) \sum_{k \in S} w_k - \left(\frac{1}{2} + \delta\right) \min_{k \in S} w_k \right] \tag{66}$$

$$\geq \frac{1}{2} \left[ \frac{\frac{3}{2} - \delta}{\frac{1}{2} + \delta} |S| - 1 \right] \left(\frac{1}{2} + \delta\right) \min_{k \in S} w_k \tag{67}$$

$$\geq \left(\frac{1}{2} + \delta\right) \min_{k \in S} w_k \tag{68}$$

For line 64 to line 65, see activation inequality. Then:

$$\frac{1}{2} \sum_{k \in \Omega_S} w_k \leq \sum_{k \in \Omega_S} w_k - \left(\frac{1}{2} + \delta\right) \min_{k \in S} w_k \tag{69}$$

$$\leq \sum_{k \in \Omega_S, w_k \geq 0} w_k - \left(\frac{1}{2} + \delta\right) \min_{k \in S} w_k \tag{70}$$

$$\leq B \tag{71}$$

The last inequality comes from the activation inequality.

$\square$

The dilation is deduced again using the duality corollary A.5.

**Proposition A.7.** *Let $S \in \Omega$ and $\left\lceil \frac{\frac{3}{2}-\delta}{\frac{1}{2}+\delta}|S| \right\rceil \geq 3$. If the BiSE of weights $W$ and $B$ is activated for dilation, then $B \leq \frac{1}{2}\sum_W w$*

*Proof.* Using corollary A.5, activated for dilation with weights $W$ and bias $B$ means activated for erosion with weights $W$ and bias $\sum_W w - B$. Therefore,

$$\frac{1}{2}\sum_W w \leq \sum_W w - B \Leftrightarrow B \leq \frac{1}{2}\sum_W w \tag{72}$$

$\square$

We show that the output is almost binary.

**Lemma A.8.**

$$L_\oplus = \max_{I \in \mathcal{I}(\delta), i \in (X_I \oplus S)^C} (I \circledast W)(i) \tag{73}$$

$$U_\oplus = \min_{I \in \mathcal{I}(\delta), i \in X_I \oplus S} (I \circledast W)(i) \tag{74}$$

*Proof.* Let

$$\mathcal{S}(a,b) := \sum_{k \in \Omega_S \setminus S} a_k w_k + \sum_{k \in S} c_k w_k \tag{75}$$

$$F_1 = \left\{ \mathcal{S}(a,b) | (a_k) \in R(\delta), (c_k) \in [0, \frac{1}{2}-\delta] \right\} \tag{76}$$

$$F_2 = \left\{ \mathcal{S}(a,b) | (a_k) \in R(\delta), (c_k) \in [0, \frac{1}{2}-\delta] \right\} \tag{77}$$

Then by definition of dilation, these sets are all the possible values taken by the convolution for pixels in or outside of the dilated binary image.

$$F_1 = \left\{ (I \circledast W)(i) | I \in \mathcal{I}(\delta), i \in (X_I \oplus S)^C \right\} \tag{78}$$

$$F_2 = \left\{ (I \circledast W)(i) | I \in \mathcal{I}(\delta), i \in X_I \oplus S \right\} \tag{79}$$

Finally, the sup and inf are reached by the values described in the proof of proposition A.2. $\square$

Lemma A.17 shows that if the BiSE with weights $W$ and bias $B$ and scaling factor $p \geq 0$ is activated for dilation. We suppose that the activation inequalities are strict. The convolution of an almost binary image $I \in \mathcal{I}(\delta)$ with weights $W$ is either: 1) below $L_\oplus$ or 2) above $U_\oplus$. Let $i \in \Omega$. In case 1),

$$I \circledast W(i) - B \leq L_\oplus - B < 0 \tag{80}$$

$$\Leftrightarrow \chi_{W,B,p}(I)(i) \leq \xi\left(p\left(L_\oplus - B\right)\right) < \frac{1}{2} \tag{81}$$

In case 2),

$$I \circledast W(i) - B \geq U_\oplus - B > 0 \tag{82}$$

$$\Leftrightarrow \chi_{W,B,p}(I)(i) \geq \xi\left(p\left(U_\oplus - B\right)\right) > \frac{1}{2} \tag{83}$$

Therefore, we have an almost binary output:

$$\delta_L := \frac{1}{2} - \xi\left(p\left(L_\oplus - B\right)\right) \tag{84}$$

$$\delta_U := \xi\left(p\left(U_\oplus - B\right)\right) - \frac{1}{2} \tag{85}$$

$$\delta_{out} := \min(\delta_L, \delta_U) \tag{86}$$

$$\chi_{W,B,p}\left(\mathcal{I}(\delta)\right) \subset \mathcal{I}(\delta_{out}) \tag{87}$$

As the bounds $L_\oplus$ and $U_\ominus$ are reached, $\delta_{out}$ is the best possible bound.

Then, for an almost binary image $I \in \mathcal{I}(\delta)$:

$$\chi_{W,B,p}(I)(i) > \frac{1}{2} \Leftrightarrow I \circledast W(i) > B \tag{88}$$

$$\Leftrightarrow i \in X_I \oplus S \tag{89}$$

$$\tag{90}$$

The same can be done with the erosion, by replacing $B$ with $\sum_W w - B$. If $p < 0$, then $\delta_L$ and $\delta_U$ becomes $-\delta_L$ and $-\delta_U$.

## A.2 Proof of proposition 3.1

First we prove for dilation.

**Proposition A.9.** *If $\chi_{\omega,\beta,p}^{\xi,W,B}$ is activated for dilation for S, then $S = \{i \in \Omega_S | w_i > \tau_\oplus\}$ with*

$$\tau_\oplus = \frac{1}{\frac{1}{2} + \delta}\left(B(\beta) - \sum_{k \in \Omega_S, W(\omega)_k < 0} W(\omega)_k\right) \tag{91}$$

*Proof.* • Let us show that $S = \left(W \leq \min_{i \in S} w_i\right)$

Let $j \in \Omega$ such that $w_j > \min_{i \in S} w_i$. Let $I = \frac{1}{2} + \delta \mathbb{1}_{\{-j\}}$. Then:

$$\left(\mathbb{1}_S(j) = \sum_{i \in \Omega_S} \mathbb{1}_X(-i)\mathbb{1}_S(i)\right) \geq 1 \Leftrightarrow 0 \in X \oplus S \tag{92}$$

$$\Leftrightarrow I \circledast W(0) > b \tag{93}$$

$$\left(I \circledast W(0) = \sum_{i \in \Omega_S} I(-i)w_i = vw_j\right) \geq \left(\frac{1}{2} + \delta\right)\min_{i \in S} w_i \tag{94}$$

$$\geq \left(\frac{1}{2} + \delta\right)\min_{i \in S} w_i + \sum_{i \in \Omega_S \setminus S, w_i \leq 0} w_i \tag{95}$$

$$> b \tag{96}$$

Then $\mathbb{1}_S(j) = 1$ and $j \in S$. Therefore, .

Let $w^* = \min_{i \in S} w_i$. Let $S_2 = \{i \in \Omega_S \mid w_i > \tau_\oplus\}$.

• Let us show that $\forall j \in \Omega_S \setminus S$ , $w_j \leq \tau_\oplus < w^*$.

The right hand side comes directly from the dilation activation property. Let $j \in \Omega_S \setminus S$. Let us reason by contradiction and suppose that $w_j > \tau_\oplus$. Then $w_j \geq vw_j \geq vw_j + \sum_{i \in \Omega_S, w_i \leq 0} w_i > b \geq \sum_{i \in \Omega_S \setminus S, w_i \geq 0} w_i \geq w_j$. We have reached a contradiction.

- Let us show that $S \subset S_2$

Let $j \in S$. Then $w_j \geq w^* > \tau_\oplus$, therefore $j \in S_2$.

- Let us show that $S_2 \subset S$

Let $j \in S_2$. If $j \notin S$, then $w_j \leq \tau_\oplus$, which is absurd. Therefore $j \in S$.

$\square$

Using corollary A.5, we can conclude for erosion as well.

**Proposition A.10.** *If $\chi_{\omega,\beta,p}^{\xi,W,B}$ is activated for erosion for $S$, then $S = \{i \in \Omega_S | w_i > \tau_\oplus\}$ with*

$$\tau_\ominus = \frac{1}{\frac{1}{2} + \delta} \Big( \sum_{k \in \Omega_S, W(\omega)_k > 0} W(\omega)_k - B(\beta) \Big) \tag{97}$$

*Proof.* If $\chi_{W,B}$ is activated for erosion for $S$, then $\chi_{W, \sum_W w - B}$ is activated for dilation for $S$. Then

$$S = \{i \in \Omega_S | W_i > \tau_\ominus\} \tag{98}$$

$$\tau_\ominus = \frac{1}{\frac{1}{2} + \delta} \Big( \sum_W w - B - \sum_{k \in \Omega_S, W(\omega)_k < 0} W(\omega)_k \Big) \tag{99}$$

$$= \frac{1}{\frac{1}{2} + \delta} \Big( \sum_{k \in \Omega_S, W(\omega)_k > 0} W(\omega)_k - B \Big) \tag{100}$$

$\square$

## A.3 Proof that $A_\psi(S)$ (eq. 16) is convex

We show it for dilation.

Let $(H, b_1)$, $(G, b_2) \in C$ and $\alpha \in ]0, 1[$.

- **Right hand side**

We have

$$\sum_{k \in \Omega_S, \alpha h_k + (1-\alpha)g_k \leq 0} h_k = \sum_{k \in \Omega_S, \alpha h_k + (1-\alpha)g_k \leq 0, h_k \geq 0} h_k + \sum_{k \in \Omega_S, \alpha h_k + (1-\alpha)g_k \leq 0, h_k \leq 0} h_k \tag{101}$$

$$\geq \sum_{k \in \Omega_S, \alpha h_k + (1-\alpha)g_k \leq 0, h_k \leq 0} h_k \tag{102}$$

$$\geq \sum_{k \in \Omega_S, h_k \leq 0} h_k \tag{103}$$

With the same reasoning,

$$\sum_{k \in \Omega_S, \alpha g_k + (1-\alpha)g_k \leq 0} g_k \geq \sum_{k \in \Omega_S, g_k \leq 0} g_k \tag{104}$$

Then,

$$\Big( \frac{1}{2} + \delta \Big) \min_{k_1 \in S, \, k_2 \in S} (\alpha h_{k_1} + (1-\alpha)g_{k_2}) \leq \Big( \frac{1}{2} + \delta \Big) \min_{k \in S} (\alpha h_k + (1-\alpha)g_k) \tag{105}$$

Therefore, by combining these two results,

$$\alpha b_1 + (1-\alpha)b_2 < \sum_{k \in \Omega_S, \alpha h_k + (1-\alpha)g_k \leq 0} (\alpha h_k + (1-\alpha)g_k) + \Big( \frac{1}{2} + \delta \Big) \min_{k \in S} (\alpha h_k + (1-\alpha)g_k) \tag{106}$$

- **Left hand side**

With the same reasoning as the right hand side, we have

$$\sum_{k\in\overline{S},\alpha h_k+(1-\alpha)g_k\geq 0} h_k \leq \sum_{k\in\overline{S},h_k\geq 0} h_k \tag{107}$$

$$\sum_{k\in\overline{S},\alpha h_k+(1-\alpha)g_k\geq 0} g_k \leq \sum_{k\in\overline{S},g_k\geq 0} h_k \tag{108}$$

$$\sum_{k\in S,\alpha h_k+(1-\alpha)g_k\geq 0} h_k \leq \sum_{k\in S,h_k\geq 0} h_k \tag{109}$$

$$\sum_{k\in S,\alpha h_k+(1-\alpha)g_k\geq 0} g_k \leq \sum_{k\in S,h_k\geq 0} g_k \tag{110}$$

Therefore,

$$\alpha b_1 + (1-\alpha)b_2 \geq$$
$$\sum_{k\in\overline{S},\alpha h_k+(1-\alpha)g_k\geq 0} (\alpha h_k+(1-\alpha g_k)) + \left(\frac{1}{2}-\delta\right) \sum_{k\in S,\alpha h_k+(1-\alpha)g_k\geq 0} (\alpha h_k+(1-\alpha g_k))$$
$$\tag{111}$$

Therefore $\alpha(H,b_1) + (1-\alpha)(G,b_2) \in C$.

For erosion, we use the duality corollary A.5 to conclude.

## A.4 Proof of proposition 3.2

We actually propose a little less powerful formulation of the proposition.

**Proposition A.11.** *Let $\mathbb{S}$ be the set of minimizers for $S$ argument for $(S,\psi) \mapsto d\left((\widehat{W},\widehat{B}), A_\psi(S)\right)$.*
*Then:*

$$\mathbb{S} \bigcap \left\{ \{s\in\Omega_S \mid \hat{w}_s \geq \hat{w}_t\} \,\Big|\, t\in\Omega_S \right\} \neq \emptyset \tag{112}$$

The previous propositions means that one of the sets defined as a thresholded set values of $\widehat{W}$ reaches the smallest distance. Therefore, it is enough to only search for these distances.

To prove this proposition, first we show the following Lemma.

**Lemma A.12.** *Let $x_1 < y_1 \in \mathbb{R}, x_2 \leq y_2 \in \mathbb{R}$. Then:*

$$(x_1-x_2)^2 + (y_1-y_2)^2 \leq (x_2-y_1)^2 + (x_1-y_2)^2 \tag{113}$$

*If $x_2 < y_2$, then*

$$(x_1-x_2)^2 + (y_1-y_2)^2 < (x_2-y_1)^2 + (x_1-y_2)^2 \tag{114}$$

*Proof.* Let $\delta_1 := y_1 - x_1 > 0$ and $\delta_2 := y_2 - x_2 \geq 0$. Then

$$(x_1-x_2)^2 + (y_1-y_2)^2 \leq (x_2-y_1)^2 + (x_1-y_2)^2 \tag{115}$$
$$\Leftrightarrow (x_1-x_2)^2 + (x_1+\delta_1-x_2-\delta_2)^2 \leq (x_2-x_1-\delta_1)^2 + (x_1-x_2-\delta_2)^2 \tag{116}$$
$$\Leftrightarrow (x_1-x_2)^2 - (x_2-x_1-\delta_1)^2 \leq (x_1-x_2-\delta_2)^2 - (x_1+\delta_1-x_2-\delta_2)^2 \tag{117}$$
$$\Leftrightarrow -\delta_1\cdot(2x_1-2x_2+\delta_1) \leq -\delta_1\cdot(2x_1-2x_2-2\delta_2+\delta_1) \tag{118}$$
$$\Leftrightarrow 0 \leq \delta_2 \tag{119}$$

If $x_2 < y_2$ is strict, then we can replace all large inequalities by strict inequalities. $\square$

Let $S^*, \psi^* \in \underset{S,\psi}{\operatorname{argmin}}\, d\left((\widehat{W},\widehat{B}), A_\psi(S)\right)$. Then

$$S^*,\psi^* \in \underset{S\subset\Omega_S,\psi\in\{\oplus,\ominus\}}{\operatorname{argmin}} \underset{(w,b)\in A_{\psi^*}(S)}{\min} \frac{1}{2}||(\widehat{W},\widehat{B})-(w,b)||_2^2 \tag{120}$$

Let $(w^*, b^*) \in \underset{(w,b) \in A_{\psi^*}(S^*)}{\mathrm{argmin}} \frac{1}{2}||(\widehat{W}, \widehat{B}) - (w, b)||_2^2$. Then:

$$S^*, \psi^*, w^*, b^* \in S \subset \Omega_S, \psi \in \{\oplus, \ominus\}, w \in \mathbb{R}^{\Omega_S}, b \in \mathbb{R}\frac{1}{2}||(\widehat{W}, \widehat{B}) - (w, b)||_2^2$$

$$\text{subject to } \begin{cases} L_{\psi^*}(w, S) - b \leq 0 \\ b - U_{\psi^*}(w, S) \leq 0 \end{cases} \tag{121}$$

Let $i, j \in \Omega_S$ such that $i \in S^*, j \notin S^*$ such that $\hat{w}_i < \hat{w}_j$. We define:

$$S_2 := S^* \cup \{j\} \setminus \{i\} \tag{122}$$

$$w_2 := k \in \Omega_S \mapsto \begin{cases} w_i^* \text{ if } k = j \\ w_j^* \text{ if } k = i \\ w_k^* \text{ else} \end{cases} \tag{123}$$

We show that $(w_2, S_2, b^*)$ results in a lesser or equal function value than $(w^*, S^*, b^*)$. First, $(w_2, S_2, b^*)$ respects the constraints: $L_{\psi^*}(w^*, S^*) = L_{\psi^*}(w_2, S_2)$ and $U_{\psi^*}(w^*, S^*) = U_{\psi^*}(w_2, S_2)$. Then we have

$$\begin{aligned} &||(\widehat{W}, \widehat{B}) - (w^*, b^*)||_2^2 - ||(\widehat{W}, \widehat{B}) - (w_2, b^*)||_2^2 \\ &= (\hat{w}_i - w_i^*)^2 + (\hat{w}_j - w_j^*)^2 - (\hat{w}_i - w_j^*)^2 - (\hat{w}_j - w_i^*)^2 \end{aligned} \tag{124}$$

Then, let us show that $w_i^* \leq w_j^*$. We reason by contradiction and suppose that $w_i^* > w_j^*$. Then

Using lemma A.12, we find that

$$||(\widehat{W}, \widehat{B}) - (w^*, b^*)||_2^2 - ||(\widehat{W}, \widehat{B}) - (w_2, b^*)||_2^2 < 0 \tag{125}$$

This contradicts the minimum hypothesis of $(w^*, b^*)$. Therefore $w_i^* \leq w_j^*$

Using again lemma A.12, we find that

$$||(\widehat{W}, \widehat{B}) - (w^*, b^*)||_2^2 - ||(\widehat{W}, \widehat{B}) - (w_2, b^*)||_2^2 \geq 0 \tag{126}$$

Therefore $S_2, \psi^* \in \underset{S, \psi}{\mathrm{argmin}} \, d\left((\widehat{W}, \widehat{B}), A_{\psi}(S)\right)$.

Therefore, for every couples $(i, j) \in S^* \cap \Omega_S \setminus S^*$, we can switch them and stay in the minimizers. By switching all of them, we reach a set of thresholded values that is also in the minimizers of the distance.

## A.5   Proof of equations 19, 20, 21 and 22, 23, 24

We suppose that $\widehat{W} \geq 0$ .The initial problem is, for $S \subset \Omega_S, \psi \in \{\oplus, \ominus\}$.

$$\underset{(w,b) \in \mathbb{R}^{\Omega_S} \times \mathbb{R}}{\mathrm{minimize}} \frac{1}{2} \sum_{i \in \Omega_S} (w_i - \hat{w}_i)^2 + \frac{1}{2}(b - \widehat{B})^2 \quad \text{subject to } \begin{cases} L_{\psi}(w, S) - b \leq 0 \\ b - U_{\psi}(w, S) \leq 0 \end{cases} \tag{127}$$

### A.5.1   Proof for dilation

We define the following problem, with constraint set $A_{\oplus}(S)$

$$\underset{(w,b) \in \mathbb{R}^{\Omega_S} \times \mathbb{R}}{\mathrm{minimize}} \frac{1}{2} \sum_{i \in \Omega_S} (w_i - \hat{w}_i)^2 + \frac{1}{2}(b - \widehat{B})^2 \quad \text{subject to } \begin{cases} \sum_{k \in \Omega_S \setminus S} w_k - b \leq 0 \\ \forall s \in S, b \leq w_s \\ \forall k \in \Omega_S \setminus S, -w_k \leq 0 \end{cases} \tag{128}$$

First we notice that $A_{\oplus}(S) \subset A(S)$. Let $W^*, B^*$ be a solution of problem 127. If $(W^*, B^*) \in A_{\oplus}(S)$, then this concludes the proof. We define

$$\mathbb{K}^- := \{k \in \Omega_S \mid w_k^* < 0\} \tag{129}$$

$$W^+ := \max(W, 0) \tag{130}$$

Then $(W^+, B) \in A_\oplus(S) \subset A(S)$ and

$$||(\widehat{W}, \widehat{B}) - (W^*, B^*)||_2^2 - ||(\widehat{W}, \widehat{B}) - (W^+, B^*)||_2^2 \leq 0 \tag{131}$$

$$\Leftrightarrow \sum_{k \in \mathbb{K}^-} \left( (w_k^* - \hat{w}_k)^2 - (0 - \hat{w}_k)^2 \right) \leq 0 \tag{132}$$

$$\Leftrightarrow \sum_{k \in \mathbb{K}^-} w_k^*(w_k^* - 2\hat{w}_k) \leq 0 \tag{133}$$

We reason by contradiction and suppose that $\mathbb{K}^- \neq \emptyset$. If $k \in \mathbb{K}^-$, then $w_k^* < 0$ and $w_k^* - 2\hat{w}_k^* < 0$ because as hypothesis, $\forall i \in \Omega, \hat{w}_i \geq 0$. If $\mathbb{K}^- \neq \emptyset$, then $\sum_{k \in \mathbb{K}^-} w_k(w_k - 2\hat{w}_k) > 0$, which is absurd. Therefore, $\mathbb{K}^- = \emptyset$.

### A.5.2 Proof for erosion

For erosion, the new problem is with constraints $A_\ominus$:

$$\underset{(w,b) \in \mathbb{R}^{\Omega_S} \times \mathbb{R}}{\text{minimize}} \quad \frac{1}{2} \sum_{i \in \Omega_S} (w_i - \widehat{w}_i)^2 + \frac{1}{2}(b - \widehat{B})^2 \quad \text{subject to} \quad \begin{cases} b - \sum_{s \in S} w_s \leq 0 \\ \forall s \in S, \sum_{i \in \Omega_S} w_i - w_s \leq b \\ \forall k \in \Omega_S \setminus S, -w_k \leq 0 \end{cases} \tag{134}$$

We notice similarly to the dilation that $A_\ominus \subset A_\psi(S)$, and the proof is the same.

### A.6 Proof of Proposition referenced in 3.2.2

We prove the following proposition.

**Proposition A.13.** *Let $S^* := \underset{S \subset \Omega_S}{argmin}\, d(\widetilde{A}(S), w)$. Then $S^* = \{i \in \Omega_S \mid w_i \geq \min_{S^*} \widehat{w}_s\}$*

Let $W \in \mathbb{R}_+^{\Omega_S}$ be non zero and $\sigma := \sum_{k \in S} w_k$. Let $L(S) := \frac{\sigma^2}{|S|}$. We first prove two lemmas.

**Lemma A.14.** *Let $S \subset \Omega$. Let $q \in \Omega_S \setminus S$. Then*

$$L(S \cup \{q\}) > L(S) \Leftrightarrow w_q > \left( \sqrt{\frac{1}{|S|} + 1} - 1 \right) \sigma \tag{135}$$

*Proof.*

$$L(S \cup \{q\}) - L(S) > 0 \Leftrightarrow \frac{(\sigma + w_q)^2}{|S| + 1} - \frac{\sigma^2}{|S|} > 0 \tag{136}$$

$$\Leftrightarrow |S|(\sigma + w_q)^2 - (|S| + 1)\sigma^2 > 0 \tag{137}$$

$$\Leftrightarrow 2w_q \cdot \sigma \cdot |S| + w_q^2|S| - \sigma^2 > 0 \tag{138}$$

This is a 2nd degree polynomial. As $|S| > 0$, the polynomial is positive outside of its two roots.

$$r_{1,2} = \frac{-2\sigma \cdot |S| \pm \sqrt{(2\sigma \cdot |S|)^2 + 4\sigma^2|S|}}{2|S|} \tag{139}$$

$$= \sigma\left( -1 \pm \sqrt{1 + \frac{1}{|S|}} \right) \tag{140}$$

By hypothesis, $w_q \geq 0$ and $\sigma \geq 0$. Therefore, we can discard the negative root and

$$L(S \cup \{q\}) - L(S) > 0 \Leftrightarrow w_q > \left( \sqrt{\frac{1}{|S|} + 1} - 1 \right) \sigma \tag{141}$$

$\square$

**Lemma A.15.** *Let $S \subset \Omega$. Let $q \in S$. Then*

$$L(S) - L(S \setminus \{q\}) > 0 \Leftrightarrow w_q > \left(1 - \sqrt{1 - \frac{1}{|S|}}\right) \cdot \sigma \tag{142}$$

*Proof.* We apply lemma A.14 to $S \setminus \{q\}$.

$$L(S) - L(S \setminus \{q\}) > 0 \Leftrightarrow w_q > \left(\sqrt{\frac{1}{|S|-1} + 1} - 1\right)(\sigma - w_q) \tag{143}$$

$$\Leftrightarrow w_q > \frac{\sqrt{\frac{1}{|S|-1} + 1} - 1}{1 + \sqrt{\frac{1}{|S|-1} + 1} - 1} \cdot \sigma \tag{144}$$

$$\Leftrightarrow w_q > \left(1 - \sqrt{1 - \frac{1}{|S|}}\right) \cdot \sigma \tag{145}$$

$\square$

Finally we prove the following proposition:

**Proposition A.16.**

$$S^* := \{k \in \Omega_S \mid w_k \geq \min_{j \in S} w_j\} \tag{146}$$

*Proof.* Let $q \in \Omega_S \setminus S^*$. Then as $S^*$ is maximal, we have $w_q \leq \left(\sqrt{\frac{1}{|S|} + 1} - 1\right)\sigma$ by A.14.

Let $j \in S$. As $S^*$ is maximal, we have $w_j > \left(1 - \sqrt{1 - \frac{1}{|S|}}\right) \cdot \sigma$ by A.15. Then

$$\left(1 - \sqrt{1 - \frac{1}{|S|}}\right) - \left(\sqrt{\frac{1}{|S|} + 1} - 1\right) = 2 - \left(\sqrt{1 - \frac{1}{|S|}} + \sqrt{1 + \frac{1}{|S|}}\right) \tag{147}$$

Then,

$$\left(\sqrt{1 - \frac{1}{|S|}} + \sqrt{1 + \frac{1}{|S|}}\right)^2 = 2 + 2 \cdot \sqrt{1 - \frac{1}{|S|^2}} \leq 2^2 \tag{148}$$

Therefore

$$1 - \sqrt{1 - \frac{1}{|S|}} \geq \sqrt{\frac{1}{|S|} + 1} - 1 \tag{149}$$

Finally,

$$\forall j \in S, \forall q \in \Omega_S \setminus S, w_j > w_q \tag{150}$$

which concludes the proof.

$\square$

**Lemma A.17.**

$$L_\oplus = \max_{I \in \mathcal{I}(\delta), i \in (X_I \oplus S)^C} (I \circledast W)(i) \tag{151}$$

$$U_\oplus = \min_{I \in \mathcal{I}(\delta), i \in X_I \oplus S} (I \circledast W)(i) \tag{152}$$

*Proof.* Let

$$S(a,b) := \sum_{k \in \Omega_S \setminus S} a_k w_k + \sum_{k \in S} c_k w_k \tag{153}$$

$$F_1 = \left\{ S(a,b) | (a_k) \in R(\delta), (c_k) \in [0, \frac{1}{2} - \delta] \right\} \tag{154}$$

$$F_2 = \left\{ S(a,b) | (a_k) \in R(\delta), (c_k) \in [0, \frac{1}{2} - \delta] \right\} \tag{155}$$

Then by definition of dilation, these sets are all the possible values taken by the convolution for pixels in or outside of the dilated binary image.

$$F_1 = \left\{ (I \circledast W)(i) | I \in \mathcal{I}(\delta), i \in (X_I \oplus S)^C \right\} \tag{156}$$

$$F_2 = \left\{ (I \circledast W)(i) | I \in \mathcal{I}(\delta), i \in X_I \oplus S \right\} \tag{157}$$

Finally, the sup and inf are reached by the values described in the proof of proposition A.2. $\qquad\square$

Lemma A.17 shows that if the BiSE with weights $W$ and bias $B$ and scaling factor $p \geq 0$ is activated for dilation. We suppose that the activation inequalities are strict. The convolution of an almost binary image $I \in \mathcal{I}(\delta)$ with weights $W$ is either: 1) below $L_\oplus$ or 2) above $U_\oplus$. Let $i \in \Omega$.
In case 1),

$$I \circledast W(i) - B \leq L_\oplus - B < 0 \tag{158}$$

$$\Leftrightarrow \chi_{W,B,p}(I)(i) \leq \xi\left( p\left( L_\oplus - B \right) \right) < \frac{1}{2} \tag{159}$$

In case 2),

$$I \circledast W(i) - B \geq U_\oplus - B > 0 \tag{160}$$

$$\Leftrightarrow \chi_{W,B,p}(I)(i) \geq \xi\left( p\left( U_\oplus - B \right) \right) > \frac{1}{2} \tag{161}$$

Therefore, we have an almost binary output:

$$\delta_L := \frac{1}{2} - \xi\left( p\left( L_\oplus - B \right) \right) \tag{162}$$

$$\delta_U := \xi\left( p\left( U_\oplus - B \right) \right) - \frac{1}{2} \tag{163}$$

$$\delta_{out} := \min(\delta_L, \delta_U) \tag{164}$$

$$\chi_{W,B,p}\left( \mathcal{I}(\delta) \right) \subset \mathcal{I}(\delta_{out}) \tag{165}$$

As the bounds $L_\oplus$ and $U_\ominus$ are reached, $\delta_{out}$ is the best possible bound.

Then, for an almost binary image $I \in \mathcal{I}(\delta)$:

$$\chi_{W,B,p}(I)(i) > \frac{1}{2} \Leftrightarrow I \circledast W(i) > B \tag{166}$$

$$\Leftrightarrow i \in X_I \oplus S \tag{167}$$

$$\tag{168}$$

The same can be done with the erosion, by replacing $B$ with $\sum_W w - B$. If $p < 0$, then $\delta_L$ and $\delta_U$ becomes $-\delta_L$ and $-\delta_U$.

# B Reparametrization Functions

To facilitate the convergence of our networks towards morphological operators, certain constraints are beneficial. In order to avoid dealing with a multitude of constraints, we reparametrize some variables to ensure that the constraints are always satisfied. We introduce two reparametrization functions for the weights, and three for the bias.

**Positive**   Our objective is to reach the set of activable weights and bias. Theorem 2.3 indicates that we only have to look at positive parameters. We can enforce them to be positive by setting $W$ and $B$ as the softplus function.

$$B(\cdot) = W(\cdot) := f^+(\cdot) := \log(1 + \exp(\cdot)). \tag{169}$$

**Dual reparametrization**   For the weights, we introduce the *dual* reparametrization as follows:

$$W_{dual}(\omega) := \frac{K \cdot f^+(\omega)}{\sum_{W(\omega)} w} \tag{170}$$

with $K := 2 \cdot \xi^{-1}(0.95)$ and $\xi$ the smooth treshold activation choosen. We can show that this dual reparametrization ensures that the training process is similar for both erosion and dilation.

Other reparametrization for the bias can be defined to keep it into coherent range values. If the bias is smaller than $\min(W)$, then $\forall X \subset \Omega, \chi(\mathbb{1}_X) < 0.5$: no values will be close to 1. On the other side, if the bias is higher than $\sum_W w$, then $\chi(\mathbb{1}_X) > 0.5$. Therefore, we want $\min(W) < B < \sum_W w$. Let $\{W_1 < ... < W_K\}$ be the ordered values taken by the weights $W$. The previous inequality is ensured if $B$ belongs to the closed convex set $C_b := [\frac{W_1 + W_2}{2}, \sum_W w - \frac{W_1}{2}] = [l_c(W), u_c(W)]$. There are two ways of ensuring that $B \in C_b$.

**Projected**   First, we can project the bias after the gradient iteration: this comes down to a projected gradient algorithm. We call this approach "Projected" reparametrization. We apply $B_p(\beta) := f^+(\beta)$, and we project $B_p(\beta)$ onto $C_b$ after the gradient update iteration.

**Projected Reparam**   The second way is to redefine $B$ before the end of the iteration, instead of after. We call this approach "Projected Reparam" reparametrization:

$$B_{pr}(\beta) := \begin{cases} l_c(W) & \text{if } f^+(\beta) < l_c(W) \\ u_c(W) & \text{if } f^+(\beta) > u_c(W) \\ f^+(\beta) & \text{else .} \end{cases} \tag{171}$$

## C  Initialization

### C.1  Initialization

In this section, we investigate the initialization of the BiSE neuron. Let us suppose that we stack $L \in \mathbb{N}^*$ BiSE neurons one after the other.

The BiSE layer performs convolutions with a smooth threshold function as the activation, resulting in values in the range of $[0, 1]$. Since we have reparametrized the weights to be positive, the classical initialization proposed in [7], which is tailored for ReLU activation and includes negative weights, needs to be adapted. We ensure that the gradients do not vanish during initialization, especially when stacking BiSE neurons in a deep network. Inspired by [7], we sample uniform weights with an adjusted distribution that maintains a constant variance.

**Notations**  Let $l \in [\![1, L]\!]$ be the layer number. Let $\mathbf{W}_l \in \mathbb{R}^{\Omega_S}$ be the random set of weights of this layer. Let $b_l \in \mathbb{R}$ be the bias (not random). Let $\mu_l := \mathbb{E}(\mathbf{W}_l)$ and $\sigma_l := Var(\mathbf{W}_l)$. Let $\mathbf{x}_0 \in \{0, 1\}^{\Omega_I}$ be the input image, and for $l \in [\![1, K]\!]$, let $\mathbf{x}_l$ be the output of the $l^{th}$ layer: $\mathbf{x}_l := \xi(\mathbf{y}_l) \in [0, 1]^{\Omega_I}$ with $\mathbf{y}_l := \mathbf{x}_{l-1} \circledast \mathbf{W}_l - b_l \in \mathbb{R}^{\Omega_I}$ such that for $j \in \Omega_I$, $\mathbf{y}_l(j) = \sum_{i \in \Omega_S} \mathbf{W}_l(i)\mathbf{x}_{l-1}(j-i) - b_l$. Let $n_l = |\mathbf{W}_l|$.

**Assumptions**

1. For $l \in [\![1, L]\!]$, the $\left(\mathbf{W}_l(i)\right)_{i \in \Omega_S^l}$ are independent, identically distributed (IID)
2. For $l \in [\![1, L]\!]$, the $\left(\mathbf{x}_l(i)\right)_{i \in \Omega_I}$ are IID
3. For $l \in [\![1, L]\!]$, $\mathbb{E}[\mathbf{y}_l] = 0$ and $p(\mathbf{y}_l)$ is symmetrical around 0
4. For $(l, i, j) \in ([\![1, L]\!] \times \Omega_S \times \Omega_I)$, $\mathbf{W}_l(i)$ and $\mathbf{x}_{l-1}(j-i)$ are independent

Under these assumptions, we can drop the $i$ and $j$ index and rewrite $\mathbf{y}_l = \sum \mathbf{W}_l \mathbf{x}_{l-1} - b_l$, with implicit indexing. The same goes for $\mathbf{x}_l$.

According to Fubini-Lebesgue theorem, and using assumption 3, we have $\mathbb{E}[\mathbf{x}_l] = \mathbb{E}[\xi(\mathbf{y}_l)] = \xi(\mathbb{E}[\mathbf{y}_l]) = 0.5$.

**Bias computation**  We have, $\forall l \in [\![1, K]\!]$, $\mathbb{E}[\mathbf{y}_l] = n_l \mathbb{E}[\mathbf{W}_l]\mathbb{E}[\mathbf{x}_{l-1}] - b_l$. We use the unbiased estimator to compute the mean, by summing all the weights of the layer: $\mu_l := \frac{1}{n_l} \sum_{w \in \mathbf{W}_l} w$. Therefore, with $\mathbb{E}(\mathbf{x}_0)$ the mean of the inputs:

$$b_1 = \mathbb{E}(\mathbf{x}_0) \sum_{w \in \mathbf{W}_1} w \tag{172}$$

$$\forall l \in [\![2, K]\!], \ b_l = \frac{1}{2} \sum_{w \in \mathbf{W}_l} w. \tag{173}$$

In some cases, we initialize the scaling factor $p = 0$ to avoid bias towards applying complementation or not. Then, this initialization leads to a point with zero grad (see Appendix C.2 for details). Therefore, we add some noise to the bias:

$$\forall l \geq 2, \ B_l = \frac{1}{2} \sum_{W_l} w + \mathcal{U}(-\epsilon_{bias}, \epsilon_{bias}) \tag{174}$$

$$B_1 = \mathbb{E}(\mathbf{x}_0) \sum_{W_1} w + \mathcal{U}(-\epsilon_{bias}, \epsilon_{bias}) \tag{175}$$

and $\epsilon_{bias}$ depends on the number of layers of the network and is set between $[10^{-4}, 10^{-2}]$.

**Variance computation**  We find a recurrence relation between $Var(\mathbf{y}_l)$ and $Var(\mathbf{y}_{l-1})$. To simplify the computations, let $p' = \frac{d\xi(p \cdot x)}{dx}(0)$ be the tangent at 0, and let us assume that $p' > 0$. We approximate the smooth threshold by a piece-wise linear function:

$$\xi(u) \simeq \left(\frac{1}{2} + p\right)\mathbb{1}_{]-\frac{1}{2p},\frac{1}{2p}[}(u) + \mathbb{1}_{]\frac{1}{2p},+\infty]}(u) \tag{176}$$

$$\mathbb{E}[\mathbf{x}_l^2] = \mathbb{E}[\xi(\mathbf{y}_l)^2] \tag{177}$$

$$\mathbb{E}[\mathbf{x}_l^2] \simeq p'^2 \int_{-\frac{1}{2p'}}^{\frac{1}{2p'}} y^2 p(\mathbf{y}_l)d\mathbf{y}_l + p' \int_{-\frac{1}{2p'}}^{\frac{1}{2p'}} \mathbf{y}_l p(\mathbf{y}_l)d\mathbf{y}_l + \frac{1}{2}\int_0^{+\infty} p(\mathbf{y}_l)d\mathbf{y}_l + \frac{1}{2}\int_{\frac{1}{2p'}}^{+\infty} p(\mathbf{y}_l)d\mathbf{y}_l \tag{178}$$

$$= p'^2 Var(\mathbf{y}_l) + H(p') + \frac{1}{4} \tag{179}$$

$$\text{with } H(p') := \int_{\frac{1}{2p'}}^{+\infty} \left(\frac{1}{2} - 2p'^2\mathbf{y}_l^2\right)p(\mathbf{y}_l)d\mathbf{y}_l. \tag{180}$$

By independence, we have

$$\frac{1}{n_l}Var(\mathbf{y}_l) = Var(\mathbf{W}_l\mathbf{x}_{l-1}) \tag{181}$$

$$= \sigma_l Var(\mathbf{x}_{l-1}) + \mathbb{E}[\mathbf{x}_{l-1}]^2\sigma_l + \mu_l^2 Var(\mathbf{x}_{l-1}) \tag{182}$$

$$= \sigma_l[\mathbb{E}[\mathbf{x}_{l-1}^2] - \mathbb{E}[\mathbf{x}_{l-1}]^2] + \mathbb{E}[\mathbf{x}_{l-1}]^2 + \mu_l^2[[\mathbb{E}[\mathbf{x}_{l-1}^2] - \mathbb{E}[\mathbf{x}_{l-1}]^2] \tag{183}$$

$$= \mathbb{E}[\mathbf{x}_{l-1}^2](\sigma_l + \mu_l^2) - \mu_l^2\mathbb{E}[\mathbf{x}_{l-1}]^2 \tag{184}$$

$$\simeq p'^2 Var(\mathbf{y}_{l-1})(\sigma_l + \mu_l^2) - \frac{1}{4}\mu_l^2 + (\sigma_l + \mu_l^2)\left(\frac{1}{4} + H(p')\right) \tag{185}$$

$$= p'^2 Var(\mathbf{y}_{l-1})(\sigma_l + \mu_l^2) + G(p') \tag{186}$$

$$\text{with } G(p') := \frac{1}{4}\sigma_l + H(p')(\sigma_l + \mu_l^2). \tag{187}$$

If $G(p') \geq 0$, which is equivalent to $H(p') \geq -\frac{\sigma_l}{4(\sigma_l+\mu_l^2)}$.

$$\frac{1}{n_l}Var(\mathbf{y}_l) \leq p'^2 Var(\mathbf{y}_{l-1})(\sigma_l + \mu_l^2). \tag{188}$$

We unroll the recursive relation. Let

$$V(p') := Var(\mathbf{y}_1) \prod_{k=1}^{l} p^2 n_k(\sigma_k + \mu_k^2). \tag{189}$$

Then

$$V(p') \leq Var(\mathbf{y}_l) \leq V(p') + \sum_{k=1}^{L}\frac{1}{4}\sigma_k \prod_{m=k}^{l} p^2 n_m(\sigma_m + \mu_m^2). \tag{190}$$

The right hand side of the inequality comes from the fact that $H(p') \leq 0$.

If $G(p') \leq 0$, which is equivalent to $H(p') \leq -\frac{\sigma_l}{4(\sigma_l+\mu_l^2)}$, we have:

$$0 \leq Var(\mathbf{y}_l) \leq V(p'). \tag{191}$$

If the variance is equal to 0, then the output is constant: therefore, the gradient is 0 and we cannot learn the parameters. On the other hand, if the variance is too high, the model is poorly conditioned: we can expect the learning to be unstable.

If the number of layers is too big, the product term $V(p')$ can either vanish or explode. To avoid variance exploding when $G(p') \geq 0$, and to avoid variance vanishing when $G(p') \leq 0$, we choose to ensure that the product is equal to 1 by imposing each term to be 1.

$$\forall l \in [\![1, L]\!] , \ p'^2 n_l(\sigma_l + \mu_l^2) = 1 \tag{192}$$

$$\sigma_l = \frac{1}{p'^2 n_l} - \mu_l^2 \tag{193}$$

$$\forall l \in [\![1, L]\!] , \ p'^2 n_l(\sigma_l + \mu_l^2) = 1. \tag{194}$$

The variance needs to be positive, which gives

$$\mu_l < \frac{1}{p'}\sqrt{\frac{1}{n_l}}. \tag{195}$$

We have a range of possible means and variances. Let us choose a uniform distribution for $\mathbf{W}_l$. Then $\mathbf{W}_k \sim \mathcal{U}(\mu_l - \sqrt{3\sigma_l}, \mu_l + \sqrt{3\sigma_l})$. We want positive weights (see theorem 2.3), therefore we want $\mu_l > \sqrt{3\sigma_l}$, which gives

$$\mu_l > \frac{\sqrt{3}}{2p'}\sqrt{\frac{1}{n_l}}. \tag{196}$$

Using equations 195 and 196, we choose $\mu_l$ as the mean of the bounds:

$$\mu_l := \frac{\sqrt{3} + 2}{4p'\sqrt{n_l}}. \tag{197}$$

**Role of the scaling factor** $p$ \quad Finally, $p'$ is related to the value of $p$: $p' = \frac{d(\xi(p \cdot x))}{dx}(0) = p\xi'(0)$. Ideally, to avoid bias towards applying complementation or not, $p$ should be set at $0$, therefore $p' = 0$. We consider that after the first few iterations, $p \neq 0$, and our computations become valid.

How to set $p'$ in practice ? We want the BiSE output to be in the full range $[0, 1]$. Let us suppose that after a few iterations, $p = 1$. With $B_l = \sum_{w \in W_l} w$, the lowest output value is given by an image full of $0$, giving $\xi(-B_l)$. The highest output is given by an image full of $1$ and gives $\xi(B_l)$. Let $h \in ]0, 1[$ (e.g. $h = 0.95$). We want $\xi(B) > h$, which is the same as $\xi(-B_l) < 1 - h$ thanks to the binary-odd property. By using the fact that $\mu_l n_l \sim \sum_{w \in W_l} w$, we have:

$$\xi(B_l) \geq h \Leftrightarrow p' \leq \frac{\sqrt{3} + 2}{8\xi^{-1}(h)}\sqrt{n_l}. \tag{198}$$

**Summary** \quad With $h = 0.95$, $\epsilon_{bias} \in [10^{-4}, 10^{-2}]$ and $p'_l := \frac{\sqrt{3}+2}{8\xi^{-1}(h)}\sqrt{n_l}$,

$$\forall l \in [\![1, L]\!] , \ \mu_l := \frac{\sqrt{3} + 2}{4p'_l\sqrt{n_l}} \tag{199}$$

$$\forall l \in [\![1, L]\!] , \ \sigma_l := \frac{1}{p'^2_l n_l} - \mu_l^2 \tag{200}$$

$$\forall l \in [\![1, L]\!] , \ \mathbf{W}_l \sim \mathcal{U}(\mu_l - \sqrt{3\sigma_l}, \mu_l + \sqrt{3\sigma_l}) \tag{201}$$

$$\forall l \in [\![1, L]\!] , \ p_l := 0 \tag{202}$$

$$\forall l \in [\![2, L]\!] , \ b_l := \frac{1}{2}\sum_{w \in \mathbf{W}_l} w + \mathcal{U}(-\epsilon_{bias}, \epsilon_{bias}) \tag{203}$$

$$b_1 := \mathbb{E}(\mathbf{x}_0)\sum_{w \in \mathbf{W}_1} w + \mathcal{U}(-\epsilon_{bias}, \epsilon_{bias}). \tag{204}$$

We can approximate $\mathbb{E}(\mathbf{x}_0)$ with the mean of a few samples, for example the first batch.

**Reparametrization** We computed the weights and biases of the convolution. In the BiSE definition 2.2, they correspond to $W(\omega)$ and $B(\beta)$. The true initialization must be done on $\omega$ and $\beta$. If $W$ and $B$ are invertible, the problem is solved:

$$\omega_l = W^{-1}(\mathbf{W}_l) \tag{205}$$

$$\beta_l = B^{-1}(b_l). \tag{206}$$

If the functions are not invertible, the study must be done case by case in order to find generative distributions for $\omega_l$ and $\beta_l$ that respects the resulting distributions for $W(\omega_l)$ and $B(\beta_l)$.

**Dual reparametrization** For the dual reparametrization, we need to adapt the parameter $K$.

**Proposition C.1.** *Let $\sigma, \mu \in \mathbb{R}_+^2$ such that $\mu - \sqrt{3\sigma} > 0$. Let $a \in \mathbb{R}_+^*$. Let $X_i \sim \mathcal{U}\left(a, a\frac{\mu+\sqrt{3\sigma}}{\mu-\sqrt{3\sigma}}\right)$ be a sequence of independent, identically distributed random variables, $S_N = \sum_{i=1}^N X_i$, $K_N = \mu \cdot N$ and $W_{i,N} = K_N \frac{X_i}{S_N}$. Then, almost surely (a.s.)*

$$W_{i,N} \longrightarrow_{N \mapsto \infty}^{a.s.} \mathcal{U}(\mu - \sqrt{3\sigma}, \mu + \sqrt{3\sigma}). \tag{207}$$

*Proof.*

$$\mathbb{E}[|X_i|] = \mathbb{E}[X_i] = \frac{a}{2}\left(1 + \frac{\mu + \sqrt{3\sigma}}{\mu - \sqrt{3\sigma}}\right) = a\frac{\mu}{\mu - \sqrt{3\sigma}} < +\infty. \tag{208}$$

Then, according to the strong law of large numbers, $\frac{S_N}{N} \longrightarrow^{a.s.} \mathbb{E}[X_i]$. Then:

$$W_{i,N} = X_i\mu\frac{N}{S_N} \longrightarrow_N^{a.s.} X_i\mu\frac{\mu - \sqrt{3\sigma}}{a\mu} = \frac{1}{a}(\mu - \sqrt{3\sigma})X_i \sim \mathcal{U}(\mu - \sqrt{3\sigma}, \mu + \sqrt{3\sigma}). \tag{209}$$

$\square$

Finally if we take $a = \mu_k - \sqrt{3\sigma_k}$ the initialization becomes:

$$K = \mu_k \cdot n_k = 2 \cdot \xi^{-1}(h) \tag{210}$$

$$\mathbf{W}_k \sim \mathcal{U}(\mu_k - \sqrt{3\sigma_k}, \mu_k + \sqrt{3\sigma_k}) \tag{211}$$

$$K\frac{\mathbf{W}_k}{\sum_{w \in W_k} w} \sim_{n_k \to +\infty}^{a.s.} \mathcal{U}(\mu_k - \sqrt{3\sigma_k}, \mu_k + \sqrt{3\sigma_k}). \tag{212}$$

**Remark on bias initialization** Theorem 2.3 expresses the operation approximated by the BiSE depending on the bias. From the second to the last layer, the bias is initialized at the middle of both operation. Therefore, the BiSE are unbiased to learn either dilation or erosion.

However, for the first BiSE layer, the bias is initialized differently. If $\mathbb{E}(\mathbf{x}_0) < \frac{1}{2}$, then the bias indicates a dilation. We bias the BiSE to learn a dilation. This is explained by the assumption that the output must mean at $\frac{1}{2}$. If the input is a lower than $\frac{1}{2}$, we must increase its value. The same goes for the erosion: if the value of the input is too high, we reduce it by applying an erosion.

## C.2 Gradient Computation

Let $\Gamma = \chi_L \circ ... \circ \chi_1$ be a sequence of BiSE neurons, with parameters $p_l$, $b_l$ and $\mathbf{W}_l$ for all layers $l$. We compute $\mathcal{L}(\Gamma(X), Y)$ for one input sample. Let

$$\Pi_l := \partial_1 \mathcal{L}\Big(\Gamma(X), Y\Big)\left(\prod_{k=l+1}^N \text{diag}(\xi'(\chi_k))p_k W_k\right). \tag{213}$$

We re-denote $\mathbf{W}_l$ as the linear matrix such that $\chi_{l-1} \circledast \mathbf{W}_l = \chi_{l-1}\mathbf{W}_l$.

$$\frac{\mathrm{d}\mathcal{L}}{\mathrm{d}p_l} := \Pi_l \mathrm{diag}(\xi'(\chi_l))(\chi_{l-1}\mathbf{W}_l - b_l) \tag{214}$$

$$\frac{\mathrm{d}\mathcal{L}}{\mathrm{d}b_l} := \Pi_l \mathrm{diag}(\xi'(\chi_l))(-p_l) \tag{215}$$

$$\frac{\mathrm{d}\mathcal{L}}{\mathrm{d}\mathbf{W}_l} := \Pi_l \mathrm{diag}(\xi'(\chi_l))p_l\phi_{l-1}. \tag{216}$$

If for all $l$, $p_l = 0$, then $\frac{\mathrm{d}\mathcal{L}}{\mathrm{d}b_l} = \frac{\mathrm{d}\mathcal{L}}{\mathrm{d}\mathbf{W}_l} = 0$. Moreover, by initialization of the bias $b_l$, we have $\chi_{l-1}\mathbf{W}_l - b_l = 0$, leading to $\frac{\mathrm{d}\mathcal{L}}{\mathrm{d}p_l} = 0$. With our current initialization, all the gradients are equal to 0. Therefore, in practice, we add a uniform noise to the bias:

$$\forall l \geq 2 \,, \; b_l = \frac{1}{2}\sum_{\mathbf{W}_l} w + \mathcal{U}(-10^{-4}, 10^{-4}) \tag{217}$$

$$B_1 = \mathbb{E}(X)\sum_{W_1} w + \mathcal{U}(-10^{-4}, 10^{-4}). \tag{218}$$

This will lead to $p_l$ moving away from 0, which unblocks the other gradients as well. We introduce a small bias towards either dilation or erosion. However, its significance is negligible, and does not prevent from learning one operation or the other.

## D    Regularization onto the set of activable parameters

We compute the distance to the closest set $A_{\psi,S}$ defined in (16) for each BiSE neuron. Let $(\mathbf{W}, b) :=$ $(W(\omega), B(\beta))$ be the weights in a given iteration. Then:

$$\mathcal{L}_{\text{morpho}} = \mathcal{L}_{acti} := \min_{S,\psi} d\bigg( (\mathbf{W}, b), A_{\psi,S} \bigg). \tag{219}$$

To do this, we must compute this distance in a differentiable fashion. We proceed in two steps: first, we compute the optimal $(S^*, \psi^*)$ as in §3.1, by checking all possible thresholded set of weights and solving the corresponding QP with OSQP, with the Lagrangian dual method. This yields the best $(\mathbf{W}^*, b^*)$ as well as the Lagrangian dual values, from which we deduce the differentiable form of the distance. More details are given in Appendix D. However, the computational burden described in §3.1 persists: the first step of computing $S^*$ is too long in practice.

If $\psi^* = \oplus$, let $\lambda^*$ be the dual value for the constraint (19) and

$$\mathbb{T} := \{t \in S^* \mid \mathbf{W}_t \le b^*\} \tag{220}$$

$$\mathbb{K} := \{k \in \Omega \setminus S^* \mid \mathbf{W}_k \le \lambda^*\} \tag{221}$$

$$\overline{\mathbb{K}} := (\Omega \setminus S^*) \setminus \mathbb{K} \tag{222}$$

$$D := |\overline{\mathbb{K}}|(|\mathbb{T}| + 1) + 1, \tag{223}$$

then we can show that we obtain the following differentiable expressions for $(\mathbf{W}^*, b^*)$.

$$b^* = \frac{1}{D}\bigg( \sum_{j \in \overline{\mathbb{K}}} \mathbf{W}_j + |\overline{\mathbb{K}}|\Big( \sum_{t \in \mathbb{T}} \mathbf{W}_t + b \Big) \bigg) \tag{224}$$

$$\forall j \in \overline{\mathbb{K}}, \mathbf{w}_j^* = \mathbf{W}_j + \frac{1}{D}\bigg( \sum_{t \in \mathbb{T}} \mathbf{W}_t + b - (|\mathbb{T}| + 1)\sum_{i \in \overline{\mathbb{K}}} \mathbf{W}_i \bigg) \tag{225}$$

$$\forall k \in \mathbb{K}, \ \mathbf{W}_k^* = 0 \tag{226}$$

$$\forall t \in \mathbb{T}, \ \mathbf{W}_t^* = b^* \tag{227}$$

$$\forall s \in S^* \setminus \mathbb{T}, \ \mathbf{W}_s^* = \mathbf{W}_s. \tag{228}$$

Then, we have a differentiable expression for the loss.

$$\mathcal{L}_{acti} = \sum_{i \in \Omega}(\mathbf{W}_i - \mathbf{w}_i^*)^2 - (b - b^*)^2. \tag{229}$$

# E  Experiments

Table 2: Best set of hyperparameters for each architecture

| Architecture | $W(\omega)$ | $B(\omega)$ | Coef Regu | Regu Delay | Learning Rate | Last Act. |
|---|---|---|---|---|---|---|
| DLUI ($W =$ Id) | Id | Id | 0 | - | $6.2 \cdot 10^{-3}$ | tanh |
| DLUI (No Regu) | $f^+$ | $f^+$ | 0 | - | $9.8 \cdot 10^{-2}$ | tanh |
| DLUI $\mathcal{L}_{exact}$ | $f^+$ | proj. rep. | 0.01 | 10000 | $4.2 \cdot 10^{-2}$ | softmax |
| DLUI $\mathcal{L}_{uni}$ | $f^+$ | proj. rep. | 0.01 | 20000 | $6.1 \cdot 10^{-2}$ | softmax |
| DLUI $\mathcal{L}_{nor}$ | $f^+$ | Id | 0.001 | 10000 | $5.4 \cdot 10^{-2}$ | softmax |

We perform a random search across a range of hyperparameters to identify the optimal configuration. The hyperparameters explored in the search are as follows:

- Learning rate between $10^{-1}$ and $10^{-3}$.
- Last activation: Softmax layer vs Normalized tanh. If Softmax Layer, we use $\mathcal{L}_{CE}$, else we use $\mathcal{L}_{BCE}$
- Applying the softplus reparametrization to the weights or not.
- The bias reparametrization schema between no reparametrization, positive, projecetd and projected reparam.
- Regularization loss: either no regularization, or the projection onto constant set (choose between $\mathcal{L}_{exact}$, $\mathcal{L}_{uni}$ and $\mathcal{L}_{nor}$). If we apply regularization, then we also apply softplus reparametrization.
- If regularization: the coefficient $c$ in the loss, either $0.01$ or $0.001$.
- If regularization: the number of batches to wait before applying the regularization, in [0, 5000, 10000, 15000, 20000].

For each regularization schema, we select the model with the best binary validation accuracy, and the corresponding results are displayed in Table 1. Detailed hyperparameter configurations are provided in Table 2 for reference.

We investigate the effects of each hyperparameter.

**Bias Reparametrization**   For the choice of bias reparametrization function, we observed that not applying any reparametrization to the bias resulted in less robustness, and the network occasionally failed to learn. Applying projected reparametrization improved robustness, but it did not increase the proportion of activated neurons. Conversely, applying the positive projected and projected reparametrization functions increased the ratio of activated BiSE neurons from 0.9% to around 20%. Ensuring that the bias falls within the correct range of values enhances the interpretability of the network.

**Regularization Delay**   when activating the regularization loss at the beginning of training, the results were notably worse, especially in binary accuracy. Our hypothesis is that the network does not have enough time to explore the right morphological operations and is prematurely drawn to a non-optimal operator, getting stuck in its vicinity. We observed that the accuracy improved when increasing the waiting time before activating the regularization loss. Future work may explore more advanced policies for applying the loss, such as using the loss at a given frequency instead of every batch, increasing the coefficient of the loss at each step, or waiting for the network to converge before applying regularization.

**Regularization coefficient and last activation**   The coefficient of the regularization loss did not have a significant impact, as well as replacing the normalized tanh by a softmax layer.

