# OpenReview forum: "A foundation for exact binarized morphological neural networks"
_NeurIPS.cc/2023/Workshop/WANT — WANT@NeurIPS 2023 Poster_

### Official Review · Reviewer_9TpA · 2023-10-23
**Review of "A foundation for exact binarized morphological neural networks"**

**Confidence:** 3

**Review:**

The authors propose a method using morpholohy to binarise convolutional neural networks without loss of performance.

The paper is well-written, and I'm mainly missing the proofs to be fully satisfied.

Major comments:
 - The proofs are missing for the theorems and propositions, which makes it difficult to assess the work. Add these.
 - Equations 31 and 32 are stated on their own without any context, and not being part of any sentence.
 - Line 414: Define he function \xi.

Minor comments:
 - When an equation is the end of a sentence, put a full stop after it.
 - Figure 2: Is of poor quality. Save the figure as a PDF instead, to maintain the full resolution.
 - Equation 27. Define d. Is this a proper distance function, or what are its properties otherwise?
 - Add a reference for the statement on line 246 to how the expert would do it.
 - Do you want to write "Figure X" or "figure X"? Be consistent. Same with "appendix", for instance.
 - Line 331: "wel" -> "well".
 - Line 455: "annexe" -> "Appendix".
 - Line 457: What is the legth of a network? Also, what is "it" in the last sentence "It is the set between" ...?
 - Equation 51: Remove the indentation.
 - Line 497: "almost certainly" -> "almost surely"? Or does certainly imply something else than surely?

---

### Official Review · Reviewer_mVvC · 2023-10-24
**The subjects discussed in the paper are beyond my expertise, nevertheless it is well written and the results look good and**

**Confidence:** 2

**Review:**

The possibility of NN binarization without performance degradation looks very attractive given the high resource utilization it grants. However, it is hard to achieve the exact binarization. Hopefully there are reparameterization methods and regularization techniques presented in the paper. But I didn't get all the math

---

### Official Review · Reviewer_ejPm · 2023-10-25
**A foundation for exact binarized morphological neural networks**

**Confidence:** 3

**Review:**

The paper presents a model based on mathematical morphology which can binarize ConvNets without loss of performance under certain, not easy to achieve, conditions. To tackle this issue, the work proposes two new approximation methods and develops a theoretical framework for ConvNets binarization using mathematical morphology. The paper also proposes regularization losses to improve the optimization.

Pros
- The current paper improves an existing BiMoNN [2] paper by tackling its limitations, namely (i) the limited aspect of the previous work to binary inputs (by providing generalization to gray-scale images and two new approximate binarization techniques to deal with the cases where exact binarization is not possible, etc), and (ii) the fact that the previous work learns one filter per layer which limits the design in modern architectures.
- The paper provides an evaluation of the binarized BiMoNN to learn complex morphological pipelines without performance loss.

Cons
- In 2.2,  it is not clear how the results discussed in that section are really enough to handle complex images, in other words, how non-binary images (or very complex ones) could be handled? according to the findings of section 2.2.
- The experimental part of the paper is limited to simple dataset and architecture.

---

### Meta-Review · Area_Chair_BRuf · 2023-10-26

**Recommendation:** Accept (Poster)
**Confidence:** 5

**Metareview:**

The paper proposes a framework based on mathematical morphology target at network binarization. The method is sound and well laid out. However, the results are notably worse than typical network binarization approaches and there is no clear path on how to scale the proposed approach to more challenging and complex datasets. That being said, I do believe that the method is interesting enough and could foster more discussion toward this direction. As per Reviewer 9TpA comments, the paper should be carefully checked for grammar errors and inconsistencies.

---

### Decision · Program_Chairs · 2023-10-28

**Decision:**

Accept (Poster)

**Comment:**

We thank the authors for their time and contribution to WANT and we are pleased to share that after the reviewing process the paper has been accepted. Congratulations! We encourage the authors to consider reviewers' feedback for the improvement of the camera-ready version. We hope to see you in person at the workshop and brainstorm on efficient training research together!